



# A lumped species approach for the simulation of secondary organic aerosol production from intermediate volatility organic compounds (IVOCs): Application to road transport in PMCAMx-iv (v1.0)

Stella E. I. Manavi[1,2] and Spyros N. Pandis[1,2]

[1]Department of Chemical Engineering, University of Patras, Patras, GR 26540, Greece
[2] Institute of Chemical Engineering Sciences, Foundation for Research and Technology-Hellas, Patras, GR 26540, Greece

*Correspondence to*: Spyros N. Pandis (spyros@chemeng.upatras.gr)

**Abstract.** Secondary organic aerosol (SOA) is formed in the atmosphere through the oxidation and condensation of organic compounds. Intermediate volatility compounds, compounds with effective saturation concentration ($C^*$) at 298 K between $10^3$ and $10^6$ µg m$^{-3}$, have high SOA yields and can be important SOA precursors. The first efforts to simulate IVOCs in chemical transport models (CTMs) used the volatility basis set (VBS), a highly parametrized scheme that oversimplifies their chemistry. In this work we propose a more detailed approach for simulating IVOCs in CTMs, treating them as lumped species that retain their chemical characteristics. Specifically, we introduce four new lumped species representing large alkanes, two lumped species representing polyaromatic hydrocarbons (PAHs) and one species representing large aromatics, all in the IVOC range. We estimate IVOC emissions from road transport using existing estimates of volatile organic compound (VOC) emissions and emission factors of individual IVOCs from experimental studies. Over the European domain, for the simulated period of May 2008, estimated IVOC emissions from road transport were about 21 Mmol d$^{-1}$, a factor of 8 higher than emissions used in previous VBS applications. The IVOC emissions from diesel vehicles were significantly higher than those from gasoline ones. SOA yields under low-NO$_x$ and high-NO$_x$ conditions for the lumped IVOC species were estimated based on recent smog chamber studies. Large cyclic alkane compounds have both high yields and high emissions, making them an important, yet understudied, class of IVOCs.

## 1. Introduction

Intermediate volatility organic compounds (IVOCs) have effective saturation concentration ($C^*$) between $10^3$ and $10^6$ µg m$^{-3}$ at 298 K and they are emitted as gases in the atmosphere (Donahue et al, 2006; Robinson et al., 2007). Despite their lower emissions compared to volatile organic compounds (VOCs), IVOCs can be important secondary organic aerosol (SOA) precursors due to their high SOA yields (Tkacik et al., 2012; Docherty et al., 2021). IVOCs include intermediate length C$_{12}$-C$_{22}$ alkanes (linear, branched, and cyclic), small polycyclic aromatic hydrocarbons (PAHs) and intermediate length aromatics. IVOC sources include diesel and gasoline vehicles (Schauer et al., 1999; Gordon et al., 2014; Zhao et al., 2014;



2015; 2016; Drozd et al., 2016; Tang et al., 2021) biomass burning (Schauer et al., 2001; Ciarelli et al., 2017b; Hatch et al.,
2017; Qian et al., 2021), ships (Huang et al., 2018; Lou et al., 2019; Su et al, 2020) and consumer products (Li et al., 2018;
Seltzer et al, 2021). There are hundreds of isomers of these relatively large compounds, so they are difficult to separate by
traditional gas chromatography-based techniques. In gas-chromatograms the majority of the emitted IVOCs usually appears
as an unresolved complex mixture (UCM) of co-eluting compounds (Schauer et al., 1999; 2001). As a result, their
identification and quantification are challenging, and their emissions are not well constrained. Usually, their emissions are
estimated based on other known emissions from the same source. Robinson et al. (2007) assumed that the IVOC emissions
in the US are 1.5 times the non-volatile primary organic aerosol (POA) emissions. This 1.5 factor is a zeroth order
assumption based on chassis dynamometer tailpipe measurements of diesel emissions (Schauer et al., 1999). A number of
follow-up studies used the same approach and the same or different ratios to POA (Murphy and Pandis, 2009; Koo et al.,
2014; Woody et al., 2016). Other studies proposed the scaling of the IVOC emissions to the corresponding VOC emissions
(Jathar et al., 2013; 2014; 2017; Akherati et al., 2019). Most of these previous efforts have assumed the same IVOC/POA or
IVOC/VOC ratio for the emissions of all sources. However, it is clear that source-specific scaling factors are needed (Lu et
al., 2018).

Until recently, the role of IVOCs in the formation of SOA was neglected as the atmospheric concentrations of
IVOCs are much lower than those of VOCs. However, IVOCs due to their size have significantly higher SOA yields than
VOCs (Lim and Ziemann, 2009; Presto et al., 2010; Tkacik et al., 2012; Docherty et al., 2021). For example, the measured
SOA yields of linear alkanes increase with increasing number of carbons (Lim and Ziemann, 2009; Tkacik et al., 2012;
Aumont et al., 2012). Besides carbon number, alkane structure also plays an important role in SOA formation. For an alkane
with a given number of carbons, the cyclic isomers have higher SOA yields compared to the linear compounds, whereas the
branched isomers have lower yields (Lim and Ziemann, 2009; Tkacik et al., 2012; Aumont et al., 2013). Smog chamber
studies with other compounds in the IVOC range, such as PAHs, have also reported high SOA yields (Chan et al., 2009;
Shakya et al., 2010; Kleindienst et al., 2012; Chen et al., 2016).

The first efforts to simulate IVOCs in chemical transport models (CTMs) used four volatility bins ($10^3$- $10^6$ µg m$^{-3}$)
of the volatility basis set (VBS) to describe their emissions and a highly parameterized chemical scheme (gas-phase reactions
with OH leading to the reduction of the volatility of the products by one or more bins compared to the precursor) to simulate
the IVOC oxidation (Robinson et al., 2007; Murphy and Pandis, 2009; Tsimpidi et al., 2010). In these schemes, each
volatility bin includes thousands of individual IVOCs that are assumed to follow the same oxidation path, even if they have
quite different chemical structure. This oversimplification of the chemistry of the IVOCs using the VBS is clearly a
weakness. A few studies used surrogate species such as naphthalene (Pye and Seinfeld, 2010) or n-pentadecane (Ots et al.,
2016) to represent all IVOCs. This allowed them to improve the description of the corresponding chemical reactions but
oversimplified the wide range of IVOCs present in the atmosphere. Jathar et al. (2014) showed that the representation of





IVOCs from combustion sources with three source-specific surrogate species resulted in improved SOA production predictions.

Zhao et al. (2014) measured the concentrations of both speciated and unspeciated atmospheric IVOCs in Pasadena, CA during the California at the Nexus of Air Quality and Climate Change (CalNex) study. They separated the UCM mass
into eleven bins ($B_{12}$-$B_{22}$) which correspond to the retention times of eleven n-alkanes ($C_{12}$-$C_{22}$). The IVOC UCM mass within each retention bin was further separated into two chemical groups one representing unspeciated branched alkanes and one representing the remaining UCM, which is likely a mixture of coeluting cyclic compounds (unspeciated cyclic compounds). These retention time bins $B_i$ will be used in the rest of our work also for linking the UCM mass with the lumped IVOCs. Using the same approach in laboratory emissions studies, Zhao et al. (2015; 2016) estimated the emission factors
(EFs) of 79 IVOCs emitted from on-road and off-road diesel and gasoline vehicles. Lu et al. (2020) simulated the SOA formation during the oxidation of these 79 IVOCs over the US, by lumping them into six species based on their volatility and their chemical characteristics. The model of Lu et al. (2020) included IVOCs emitted from mobile sources, and it was extended by Qin et al. (2021) and Pennington et al. (2021) to include IVOCs emitted from consumer products. The emission factors reported by Zhao et al. (2015; 2016) have also been used to estimate the IVOC VBS emissions from diesel and
gasoline vehicles in the Po Valley (Northern Italy) (Giani et al., 2019). Other studies have followed the Zhao et al. (2014) approach to estimate emissions both from on-road transport (Tang et al., 2021; Fang et al., 2021) and from various other sources such as non-road construction machinery (Qi et al., 2019), residential solid fuel combustion (Qian et al., 2021) and ship engines (Huang et al., 2018; Lou et al., 2019; Su et al., 2020). Li et al. (2019) conducted field measurements in the city of Shanghai in China and characterized IVOCs by using the approach of Zhao et al. (2014).

Improving the simulation of the formation of SOA from IVOCs (SOA-iv) in CTMs could reduce the gap between measured and predicted SOA (Pye and Seinfeld, 2010; Barsanti et al., 2013; Jathar et al., 2014; Ots et al., 2016; Zhao et al., 2016; Giani et al., 2019; Lu et al., 2020). In this work, we develop a new approach for simulating IVOCs in atmospheric CTMs treating IVOCs as lumped species that retain their chemical characteristics (alkanes, alkenes, aromatics, polyaromatics, etc.). Their atmospheric chemistry and resulting SOA formation is described similarly to that of the larger
VOCs. The proposed lumping scheme, the source specific emissions, the lumped chemical mechanism, and SOA formation parametrization are described here. The implementation of the new IVOC approach in PMCAMx (the new version is called PMCAMx-iv), a three-dimensional CTM, and its evaluation is described in a subsequent publication. The proposed approach is general enough to be portable to other regional and global CTMs. In this work, the proposed IVOC scheme is applied on on-road transportation and more specifically on IVOCs emitted by diesel and gasoline vehicles following the studies of Zhao
et al. (2015; 2016). IVOCs from other sources will be the topic of future work.





## 2. Methods

### 2.1 The SAPRC gas-phase mechanism

The gas-phase chemical mechanism employed in this application is based on the SAPRC mechanism (Carter, 2010; Environ., 2013). Most of the VOCs are simulated in the SAPRC as lumped species. The criteria used to lump the individual

VOCs are their chemical characteristics and their reaction rate constant with the hydroxyl radical ($k_{OH}$). The version of SARPC used as the starting point of this approach includes 237 reactions of 91 gases and 18 free radicals. Five lumped species represent alkanes (ALK1, ALK2, ALK3, ALK4, ALK5), two represent olefins (OLE1, OLE2), two represent aromatics (ARO1, ARO2), and there is also one monoterpene (TERP) and one sesquiterpenes species (SESQ).

### 2.2 The current VBS approach in PMCAMx

PMCAMx treats both primary and secondary organic species as chemically reactive using the VBS approach (Donahue at al., 2006). Here we present a brief summary of the VBS approach, while detailed information about its implementation to PMCAMx is provided by Murphy and Pandis (2009) and Tsimpidi et al. (2010). In the original VBS approach, OA is discretized into 9 logarithmically spaced bins characterized by their effective saturation concentration at 298 K ($C^*$ equal to 0.01-$10^6$ μg m$^{-3}$). IVOCs occupy the four highest VBS-bins ($C^*$ equal to $10^3$, $10^4$, $10^5$ and $10^6$ μg m$^{-3}$), they are emitted in the

gas phase and can form SOA-iv as they react with the hydroxyl radical. The aging OH reactions have a reaction rate constant of $4 \times 10^{-11}$ cm$^3$ molecule$^{-1}$ s$^{-1}$. Each reaction reduces the volatility of the oxidized vapor product by one order of magnitude, and it increases the mass by 7.5% to account for the added oxygen. The following reactions describe the SOA-iv formation in the VBS approach currently in use:

IVOC$_i$ (g) + OH → 1.075 O-IVOC$_{i-1}$ (g)            (R1)

O-IVOC$_{i-1}$ (g) ↔ SOA-iv $_{i-1}$ (p)            (R2)

where $i$ is the corresponding volatility bin, O-IVOC$_i$ the secondary gas phase products from the oxidation of IVOCs and SOA-iv$_{i-1}$ the aerosol products from the oxidation of the IVOC precursors.

### 2.3 The new IVOC lumping scheme

In the new IVOC modelling scheme, seven new lumped species are added to the SAPRC mechanism to describe the IVOCs

based on their chemical type and their reaction rate constant with the hydroxyl radical ($k_{OH}$). This is consistent with the approach that has been used to develop the rest of the SAPRC mechanism. Four species (ALK6, ALK7, ALK8, ALK9) are used to represent C$_{12}$-C$_{22}$ alkanes. One lumped species (ARO3) is used to represent aromatics with carbon numbers from 11 to 22 and two species (PAH1, PAH2) to represent C$_{10}$-C$_{17}$ PAHs. The new lumped species and their components are depicted in Table 1.



The individual compounds lumped to the seven new species are based on the studies of Zhao et al. (2015; 2016). The unspeciated cyclic compounds in the $B_{12}$-$B_{16}$ retention bins are dominated by aliphatic compounds in diesel engine emissions and by aromatic compounds when they are emitted by gasoline vehicles (Zhao et al., 2015; 2016). The new lumped alkanes contain speciated and unspeciated linear, branched, and cyclic alkanes. ALK6 includes alkanes and other non-aromatic compounds that react only with the hydroxyl radical and have a $k_{OH}$ between 1.3 and $1.8 \times 10^{-11}$ cm$^3$ molecule$^{-1}$
s$^{-1}$, this corresponds to linear alkanes with 12 to 14 carbons. ALK7 includes linear, cyclic and branched $C_{15}$-$C_{17}$ alkanes that have a $k_{OH}$ between 1.8 and $2.2 \times 10^{-11}$ cm$^3$ molecule$^{-1}$ s$^{-1}$. Speciated and unspeciated $C_{18}$-$C_{20}$ alkanes with a $k_{OH}$ between 2.2 and $2.6 \times 10^{-11}$ cm$^3$ molecule$^{-1}$ s$^{-1}$ are lumped in ALK8, whereas ALK9 includes alkanes and other non-aromatic compounds with a $k_{OH}$ greater than $2.6 \times 10^{-11}$ cm$^3$ molecule$^{-1}$ s$^{-1}$. The new aromatic species ARO3 contains speciated alkylbenzenes that are not explicitly represented in the original SAPRC mechanism. The new PAH species contain unsubstituted and substituted
PAHs and unspeciated larger aromatic compounds. PAH1 includes PAH compounds, such as naphthalene and methylnaphthalene isomers that have $k_{OH}$ smaller than $7 \times 10^{-11}$ cm$^3$ molecule$^{-1}$ s$^{-1}$. PAH2 includes PAH compounds, such as acenaphthylene and acenaphthene that have a $k_{OH}$ greater than $7 \times 10^{-11}$ cm$^3$ molecule$^{-1}$ s$^{-1}$.

        In the original SAPRC mechanism, ALK5 represented larger alkanes and other non-aromatic compounds that react with the hydroxyl radical with a $k_{OH}$ greater than $0.6 \times 10^{-11}$ cm$^3$ molecule$^{-1}$ s$^{-1}$. To avoid double counting in the new scheme,
ALK5 now includes compounds with a $k_{OH}$ between 0.6 and $1.3 \times 10^{-11}$ cm$^3$ molecule$^{-1}$ s$^{-1}$. In the revised ALK5, the largest compounds represented are undecane and its isomers. The aromatic compounds lumped in the ARO3, PAH1 and PAH2 were not represented in the original SAPRC mechanism and thus no change is needed for the ARO2 species.

        For the speciated individual compounds, the OH reaction rate constants are taken from the literature when available (Ananthula et al., 2006; Atkinson and Arey, 2003; Lee et al., 2003; Phousongphouang and Arey, 2002; Reisen and Arey,
2002; Kwok et al., 1997) or estimated using structure-reactivity relationships (Kwok and Atkinson, 1995; Kameda et al., 2013; Zhao et al., 2015) or extrapolated from similar compounds. For the OH reaction constants of the unspeciated alkanes, the approach of Zhao et al. (2014) was adopted. Specifically, for the unspeciated alkanes in the $n$th bin, Zhao et al. (2014) assumed that their $k_{OH}$ is the same as the linear alkane with the same number of carbons as the number of the retention time bin, $B_i$. This assumption was used for estimating the values of both unspeciated branched and cyclic alkanes in each retention
bin. For example, the unspeciated cyclic alkanes in the $B_{13}$ bin were assumed to have an OH reaction rate constant equal to that of n-tridecane ($C_{13}H_{28}$). These assumed OH reaction rate constants are a lower bound as the branched and cyclic isomers of an n-alkane have higher rate constants compared to the linear n-alkane. The OH reaction rate constants of the unspeciated aromatic compounds in the $B_{12}$-$B_{16}$ retention bins are assumed to be equal to those of naphthalene, methylnaphthalene, dimethyl naphthalene, trimethyl naphthalene and tetramethyl naphthalene respectively following the approach proposed by
Zhao et al. (2016).

        The molecular weights ($MW$) of the speciated and unspeciated individual compounds are depicted together with the OH reaction rate constants of the compounds in Table 1. The $MW$ of both unspeciated branched and cyclic alkanes in the $n$th





retention time bin are assumed to be approximately equal to that of the corresponding linear n-alkane. This may result in a minor underestimation of the molecular weight of some of the branched or cyclic alkanes (Zhao et al., 2014). For the *MW* of

the unspeciated aromatic compounds lumped in the new PAH species, the *MW* values of naphthalene and methylnaphthalene isomers are used as surrogates. The *MW* of the new lumped species are estimated as an emissions weighted average of the *MW* of the individual compounds lumped in each species (Table 1).

The effective saturation concentrations ($C^*$) of the individual compounds (Table 1) are mainly needed for comparisons with the VBS approach and they are not an essential part of the proposed approach. For the speciated

compounds, $C^*$ was calculated as:

$$C_i^* = \frac{\zeta_i \, 10^6 \, P_i^{vap} \, MW_i}{R \, T} \tag{1}$$

where, $C_i^*$ (in µg m$^{-3}$) is the effective saturation concentration for the individual compound $i$, $\zeta_i$ is its activity coefficient (assumed to be equal to unity for all the compounds), $P_i^{vap}$ is its saturation vapor pressure, $MW_i$ its molecular weight, $T$ is the temperature and $R$ is the ideal gas constant. The Nannoolal et al. (2018) group contribution method was used to estimate the

vapor pressure for each compound. In order to be able to compare the new lumped species with the IVOCs of the VBS approach, the few compounds with $C^*$ bellow $10^3$ µg m$^{-3}$ are assumed to be in the $10^3$ µg m$^{-3}$ volatility bin and the few compounds with $C^*$ above $10^6$ µg m$^{-3}$ are placed in the $10^6$ µg m$^{-3}$ bin.

### 2.4 Volatile, semi-volatile and low volatility products from the oxidation of the new lumped IVOCs

Each of the seven new lumped IVOC species reacts with the hydroxyl radical and produces both volatile products, that

remain in the gas phase, and less volatile products that can partition to the aerosol phase forming SOA-iv. In the new approach, the volatile products are simulated explicitly following the SAPRC framework, while the traditional VBS scheme is used to simulate the semi-volatile and low volatility products of the IVOCs. Specifically, we assume that each of the seven newly added reactions with the hydroxyl radical contributes to the formation of the same five products with effective saturation concentrations of 0.1, 1, 10, 100, and $10^3$ µg m$^{-3}$ at 298 K. These products then partition between the gas and the

particulate phase, forming SOA-iv. For the more volatile products of the reactions, we have assumed that they are the same as the ones produced from the hydroxyl radical reactions of the larger lumped VOCs that are already present in the SAPRC mechanism and have similar chemical characteristics.

The volatile products of the reactions of the four new lumped alkanes are assumed as a zeroth approximation to be the same as the ones produced by the reaction of ALK5 with the hydroxyl radical. As an example, the reaction of ALK6 with

the hydroxyl radical is:

**ALK6** + **OH** → 0.653 RO2R + 0.347 RO2N + 0.948 R2O2 + 0.026 HCHO + 0.099 CCHO + 0.204 RCHO + 0.072 ACET + 0.089 MEK + 0.417 PROD + $\sum_{i}^{n=5} a_i OCG_i$ \hfill (R3)





where, RO2R is the organic peroxy radical converting NO to $NO_2$ with $HO_2$ production, RO2N is the organic peroxy radical converting NO to organic nitrate, R2O2 is the organic peroxy radical converting NO to $NO_2$, HCHO is formaldehyde, CCHO

is acetaldehyde, RCHO represents the higher aldehydes (based on propionaldehyde), ACET is acetone, MEK is methyl ketone, PROD represents other organic products, $OCG_i$ is the $i$th oxygenated condensable compound which can partition to the aerosol phase, and $a_i$ is its $NO_x$-dependent mass-based yield. The reaction rate constant for the reaction R3 is assumed conservatively to have the value of $1.4 \times 10^4$ ppm$^{-1}$ min$^{-1}$; the same as the reaction of ALK5 with the hydroxyl radical. A similar reaction is used for the other new alkanes (ALK7-ALK9).

For the new aromatic and PAH species in the IVOC range, volatile products produced by the reaction with the hydroxyl radical are assumed as a zeroth approximation to be the same as the ones produced by the corresponding reaction of ARO2. As an example, the reaction of ARO3 with the hydroxyl radical is:

**ARO3 + OH** $\rightarrow$ 0.187 HO2 + 0.804 RO2R + 0.009 RO2N + 0.097 GLY + 0.287 MGLY + 0.087 BACL + 0.187 CRES + 0.05 BALD + 0.561 DCB1+ 0.099 DCB2 + 0.093 DCB3 + $\sum_{i}^{n=5} a_i OCG_i$         (R4)

where, HO2 is the hydroperoxyl radical, GLY is glyoxal, MGLY is methylglyoxal, BACL is biacetyl, CRES is cresol, BALD is benzaldehyde and DCB1-DCB3 represent three different aromatic ring opening dicarbonyl products. The reaction rate constant for the reaction R4 is assumed to have the value of $3.9 \times 10^4$ ppm$^{-1}$ min$^{-1}$; the value of the $k_{OH}$ for the reaction of ARO2 with the hydroxyl radical. Similar reactions are assumed for PAH1 and PAH2.

### 2.4.1 Estimating the yields for the new lumped IVOCs

For the simulation of the produced SOA-iv in the new scheme, it is necessary to estimate the $NO_x$ dependent mass-based yields ($a_i$) for each of the new lumped IVOC species. The first step of this process includes creating a database with smog chamber measurements of the SOA yields ($Y$) of the individual compounds at different organic aerosol concentrations ($C_{OA}$). Then, a fitting algorithm was used to estimate the yields for each of the studied precursors. The fitting algorithm estimates five mass-based coefficients ($a_i$); one for each of the five VBS products with $C^*$ equal from 0.1 to $10^3$ µg m$^{-3}$. Moreover,

when there are enough experimental data in the literature, different $a_i$'s are estimated under high and low $NO_x$ conditions. In the final step of this process, the five $NO_x$ dependent $a_i$'s of the seven new lumped IVOC species are determined as a mass emissions weighted average of the estimated $a_i$'s of the individual compounds lumped in each new species. In this study, as weights we utilize the mass-based fractions of the individual compound emissions coming from on-road diesel and gasoline vehicles (Table S1).

### 2.4.2 Fitting algorithm for estimating the mass-based yields

The fitting algorithm of Stanier et al. (2008) is utilized in this work to estimate the yields $a_i$ for the individual compounds for fixed $C_i^*$ corresponding to the VBS bins. Five SOA products with $C_i^*$ at 298 K of 0.1, 1, 10, 100 and $10^3$ µg m$^{-3}$ were chosen. The algorithm estimates the values of the $a_i$'s and of the effective vaporization enthalpy ($\Delta H$) to reproduce the SOA



measurements assuming the formation of a pseudo-ideal organic solution in the particulate phase. The algorithm tries to
minimize the following objective function $Q$:

$$Q = \sum_i \left[ Y_{i,meas} - Y_{i,pred}(a_i, \Delta H) \right]^2 \qquad (2)$$

where $Y_{i,meas}$ are the measured aerosol SOA yields and $Y_{i,pred}$ is the corresponding predicted yield for the choices of the
parameters, using the VBS framework. The objective function $Q$ is minimized by using the *fmincon* MATLAB function
(MathWorks, 2020). By minimizing the objective function, the optimal $\Delta H$ and $a_i$'s are determined for the chosen $C_i^*$ basis
set.

### 2.4.3 SOA yield measurements for individual IVOCs

The experimental studies used to estimate the individual compounds' mass-based yields with the algorithm described above
are summarized in Table 2. For individual speciated alkanes in the IVOC range, smog chamber studies have focused on
linear alkanes with available data covering only the high $NO_x$ conditions (Lim and Ziemann, 2009; Presto et al., 2010;
Docherty et al., 2021). The studied linear alkanes include: n-dodecane ($C_{12}H_{26}$), n-tridecane ($C_{13}H_{28}$) and n-tetradecane
($C_{14}H_{30}$) which are lumped in ALK6 and n-pentadecane ($C_{15}H_{32}$), n-hexadecane ($C_{16}H_{34}$) and n-heptadecane ($C_{17}H_{36}$) which
are lumped in ALK7. The SOA production during the photo-oxidation of naphthalene, 1-methylnaphthalene and 2-
methylnaphthalene has been investigated in a series of studies (Chan et al., 2009; Shakya et al., 2010; Kleindienst et al.,
2012; Chen et al., 2016). In these studies, the SOA yields of PAH's have been determined under both high and low $NO_x$
conditions. For the individual compounds lumped in ALK7, ALK8, PAH2 and ARO3 there were no experimental data.

For the individual compounds with available data, the estimated mass-based yields are based on an assumed organic
aerosol density, which equals to 1 g cm$^{-3}$ for the linear alkanes, 1.5 g cm$^{-3}$ for naphthalene, 1.4 g cm$^{-3}$ for 1-
methylnaphthalene and 1.3 g cm$^{-3}$ for 2-methylnaphthalene (Lim and Ziemann, 2009; Presto et al., 2010; Chan et al., 2009;
Shakya et al., 2010; Chen et al., 2016). For the new lumped alkanes (ALK6-ALK9), their estimated mass-based yields are
based on an assumed aerosol density of 1 g cm$^{-3}$, which is equal to that of the linear alkane SOA. For both PAH1 and PAH2,
the assumed aerosol density is equal to 1.3 g cm$^{-3}$, which corresponds to the aerosol density assumed for 2-
methylnaphthalene. In the case of ARO3, the density assumed is 1 g cm$^{-3}$, which is equal to that of ARO2 SOA.

IVOCs that have been studied so far represent only a small fraction of the total IVOC emissions. N-dodecane, n-
tridecane and n-tetradecane compounds are only 5% of the total ALK6 emissions whereas n-pentadecane, n-hexadecane and
n-heptadecane compounds represent only 4% of the total ALK7 emissions. Naphthalene, 1-methylnaphthalene and 2-
methylnaphthalene compounds represent 19% of the total PAH1 emissions. Several assumptions were applied to compensate
for the missing information. In the future, as more experimental data become available, the assumptions described below can
be relaxed.

In order to estimate the missing mass-based yields of speciated (linear, branched and cyclic) and unspeciated
(branched and cyclic) alkanes we adopted the approach of Zhao et al. (2014). For the missing speciated n-alkanes, we use the



mass-based yields of n-heptadecane, the highest n-alkane for which there were data in the literature. This is a conservative assumption as the number of carbons of linear alkanes increases so does the aerosol mass fraction of the species (Lim and Ziemann, 2009; Presto et al., 2010; Aumont et al., 2012). For the missing speciated branched alkanes, we use the mass-based yields of the n-alkane with the same carbon minus the branching methyl groups. For example, the mass-based yields of 2,6,10-trimethyltridecane are assumed to be the same as these of n-tridecane. This provides a lower bound for our estimations, as it has been suggested that the SOA yields decrease as the number of branching methyl groups increase (Lim and Ziemann, 2009; Tkacik et al., 2012; Aumont et al., 2012). For the missing speciated cyclic alkanes, we use the mass-based yields of the n-alkane with the same number of carbons. This is again a lower bound estimate as experimental studies have shown that for a given carbon number the cyclic isomers have higher SOA yields compared to those of the linear alkanes (Lim and Ziemann, 2009; Tkacik et al., 2012). For the unspeciated branched-alkanes, the mass-based yields are assumed to be the same as those of the linear n-alkane with equal number of carbons minus two. This assumption accounts for the effects of branching assuming that on average the unspeciated branched alkanes have 4 methyl branches. For the unspeciated cyclic alkanes, the mass-based yields are assumed to be the same as those of the linear n-alkane with equal number of carbons. All the surrogate species utilized in each case can be found in Table S2.

The three PAHs, for which we have sufficient data to estimate the $a_i$ yield parameters, are lumped in PAH1. For the unspeciated aromatic compounds in the $B_{12}$ retention bin, the estimated naphthalene mass-based yields are used, following the approach of Zhao et al. (2016). In order to compensate for the missing data of the other compounds which are lumped in PAH1 and PAH2, the 2-methylnaphthalene mass-based yields are used (Table S2).

Finally, since there is no information about the SOA formation of the alkylbenzenes which are lumped in ARO3, we assumed that their mass-based yields were 20% higher than those of ARO2, the lumped species that already exists in the mechanism and that represents aromatic compounds (xylene and trimethyl benzene isomers). This assumption is a starting point for the new aromatic lumped species, and it will be evaluated in a subsequent publication.

## 2.5 Estimating the road transport emissions of the new IVOC lumped species

In this first application of the new scheme, we focus on IVOCs from diesel and gasoline vehicles, but the methodology described below can be easily used for other sources. The emissions of the individual IVOCs are estimated by combing source specific emission factors (EFs) of the individual compounds with source specific EFs of VOCs. The emission rates of the individual IVOC $i$ from the source $j$ ($E_{i,j}$ in g h$^{-1}$) is estimated as:

$$E_{i,j} = \frac{EF_{i,j}}{EF_{VOC,j}} E_{VOC,j} \tag{3}$$

where, $EF_{i,j}$ (g kg$_{fuel}$$^{-1}$) is the $j$ source emission factor of the $i$th IVOC, $EF_{VOC,j}$ (g kg$_{fuel}$$^{-1}$) the source specific emission factor of the total VOCs emitted from source $j$ and $E_{VOC,j}$ (g h$^{-1}$) the emission rates of the total VOCs emitted by source $j$. The emissions of new lumped species emitted by each source, are estimated by adding all the individual compound emissions:





$$EM_{k,j} = \sum_{i=1}^{n} E_{i,j} \tag{4}$$

where $EM_{k,j}$ (g h$^{-1}$) is the emission rate of the lumped species $k$, which contains n individual compounds, by source $j$.

The EFs of individual IVOCs from gasoline vehicles are based on the measurements of emissions of light duty
gasoline vehicles (LDGVs) by Zhao et al. (2016) in the US. The application of these emission factors to European vehicles is
a necessary assumption at this stage. The Zhao et al. (2016) study was based on a versatile fleet of gasoline vehicles under
different driving cycles and the corresponding factors should be a zeroth approximation for European cars. The EFs of
individual IVOCs from diesel vehicles used here are based on Zhao et. (2015), who studied a combination of heavy-duty
(HDDVs) and medium duty diesel vehicles (MDDVs) used in the US. Due to differences in regulations, in the US passenger
cars with diesel engines account only for 3% of the vehicles in circulation (Chambers and Schmitt, 2015), whereas in
Europe, for 42.3% (EUROSTAT, 2020). For the purposes of this work, the EFs of individual IVOCs from diesel vehicles are
assumed to be the same as those of the HDDVs in Zhao et al. (2015). This assumption probably leads to an overestimation of
the diesel vehicle emissions, although previous studies have found that IVOC emissions depend more on fuel types rather
than the type of vehicles (Cross et al., 2015; Lu et al., 2018). The emissions factors of the individual compounds from diesel
and gasoline vehicles are summarized in Table S1. The EFs of the total emitted by the LDGVs and the HDDVs are taken
respectively from the studies of Zhao et al. (2016) and Zhao et al. (2015). Finally, the total VOC emissions are based on the
GEMS emissions inventory (Visschedijk et al., 2007).

## 3. Results and Discussion

### 3.1 Estimated road transport emissions of IVOCs over Europe

The domain for the applications of this work covers a region of $5400 \times 5832$ km$^2$ over Europe, with a $36 \times 36$ grid
resolution and 14 vertical layers of 6 km. The period used is that of May 2008 which corresponds to the EUCAARI intensive
measurement campaign across Europe. Using the old VBS approach for estimating the IVOC emissions, for the simulated
period, 17% of the total anthropogenic IVOC emissions were attributed to on-road transport, of which 57% was emitted by
diesel vehicles, 21% by gasoline engines and 22% was due to non-exhaust emissions from vehicles.
The spatially and temporally resolved emissions from on-road diesel and gasoline vehicles of all the individual
organic compounds depicted in Table 1, were estimated for the simulated period. As an example, the average emissions of n-
dodecane and the corresponding emissions of ALK6, the lumped species that includes n-dodecane, are shown in Figure 1.
The estimated average transportation emissions of n-dodecane in different areas of Europe range from zero to 3.6 mol d$^{-1}$
km$^{-2}$. The highest emissions of n-dodecane are located in major European cities, such as Athens, Paris, Madrid and London
and regionally in countries like Italy, Netherlands, UK, Poland, etc. For ALK6 estimations of the average transport emissions
range from zero to 182.3 mol d$^{-1}$ km$^{-2}$. Again, the highest values of the ALK6 emissions are located in major European cities.





The transportation emissions of ALK6 are higher than those of n-dodecane, as n-dodecane corresponds to only 2% of the ALK6 emissions (Table S1). The spatial distributions of the emissions of most IVOCs are quite similar to this example.

The estimated total on-road transportation emissions of the major individual compounds lumped into ALK6 are shown in Figure 2. For both gasoline and diesel vehicles, the estimated emissions of the unspeciated alkanes are higher compared to the emissions of the other compounds lumped in ALK6. The unspeciated cyclic alkanes represent 73% of the total emitted ALK6 mass and the unspeciated branched alkanes are another 21%. The three speciated linear alkanes (n-dodecane, n-tridecane and n-tetradecane) contribute 5% and the other components less than 1% each. The unspeciated cyclic alkanes in the $B_{13}$ retention bin are the predominant components of the ALK6 diesel emissions. This is consistent with the

measurements of Gentner et al. (2012) who reported that diesel emissions are dominated by aliphatic compounds with 13 to 18 carbons. The mass of the unspeciated cyclic compounds in the $B_{12}$-$B_{14}$ retention time bins, which appears to be missing from the gasoline ALK6 emissions (Figure 2), is lumped in PAH1 as it is of aromatic nature (Zhao et al., 2016). The total estimated emissions of the individual compounds in PAH1 from both sources are shown in Figure 3. Again, the unspeciated compounds are estimated to be the most important contributors to the total PAH1 emissions (75% of the total PAH1),

followed by naphthalene and the methylnaphthalene isomers (19% of the total PAH1). The unspeciated cyclic aromatic compounds in the $B_{12}$ retention bin have the highest total gasoline emissions, contributing 50% of the total PAH1 emissions from gasoline vehicles. This is consistent with the measurements of Drozd et al. (2019) who reported that for gasoline vehicles, aromatic compounds in the $B_{12}$ and $B_{13}$ retention time bins have the highest IVOC emissions.

    Using the new approach, the total IVOC emissions from diesel vehicles in Europe are approximately 16,500 kmol

d$^{-1}$, and from gasoline vehicles 4,500 kmol d$^{-1}$ (Figure 4). Previous studies have produced qualitatively consistent results with this estimation; diesel vehicles emit compounds predominantly in the IVOC range, while gasoline vehicles emit both VOCs and IVOCs (Gentner et al., 2012). The new total IVOCs emissions from on-road transportation are estimated to be 8 times higher than those calculated using the old VBS-approach. The total VBS emissions of IVOCs from diesel vehicles were 1,950 kmol d$^{-1}$ and from gasoline vehicles 690 kmol d$^{-1}$. The higher IVOC emissions with the new approach compared

to the VBS are consistent with the analysis of Lu et al. (2018). These authors reported that estimating IVOC emissions by multiplying POA emissions with the 1.5 factor (current VBS approach) underestimates the IVOC emissions observed in experimental studies.

    An advantage of the new lumped species' approach is that the IVOC emissions retain their chemical characteristics. We estimate that 98% of the diesel IVOC emissions are large alkanes (ALK6 – ALK9) and 76% of the gasoline IVOC

emissions are PAHs (PAH1 and PAH2) (Figure 4). These are mainly unspeciated compounds (unspeciated branched and cyclic alkanes and unspeciated aromatic compounds). The most important contributor to the total IVOC emissions from diesel vehicles is ALK6 (7,680 kmol d$^{-1}$), followed by ALK7 (4,900 kmol d$^{-1}$) and ALK8 with total emissions of 2900 kmol d$^{-1}$. For gasoline vehicles, the highest emissions are again estimated to be those of PAH1 (3890 kmol d$^{-1}$) followed by ALK6





which contributes 13% of the total new gasoline emissions. Aromatic and poly-aromatic compounds are more prominent in
gasoline emissions (Lu et al., 2018; 2020).

In order to better compare the new and old IVOC emissions we have estimated the volatility distribution of the
former. The volatility distribution of the emissions is quite different for the diesel emissions (Figure 5). The traditionally
assumed VBS diesel IVOC emissions are increasing as the volatility is increasing. Using the new lumped species approach
the IVOCs with $C^* = 10^5$ μg m$^{-3}$ have the highest emissions representing 43% of the total. The new distribution resembles
more closely the measured volatility distribution of the unburned diesel fuel (Lu et al., 2018; Drozd et al., 2019).

The old and new volatility distributions for the on-road gasoline vehicle are shown in Figure 6. In this case, the
highest emissions are those of the more volatile IVOCs with $C^* = 10^6$ μg m$^{-3}$. In the lumped species approach 83% of the
total IVOCs emitted from gasoline vehicles have a saturation concentration of $10^6$ μg m$^{-3}$, while with the traditional VBS
approach 43% of the emissions have the same volatility. Once more, the volatility distribution of the new emissions more
closely resembles that of the unburned fuel (Lu et al., 2018).

### 3.2 SOA yields for the new lumped IVOC species

### 3.2.1 Large alkane yields

The yields of the linear alkanes with 10 to 17 carbons were estimated based on experimental data. The estimated parameters
for all the speciated straight alkanes can be found in Table S3. As an example, the heptadecane yield as a function of the OA
concentration is shown in Figure 7. At room temperature, the estimated SOA yield at OA concentration of 1 μg m$^{-3}$ is 14%
and at 10 μg m$^{-3}$ it is 42.6%. Organic aerosol concentrations of 1-10 μg m$^{-3}$ are often encountered in the atmosphere so
heptadecane is a good SOA precursor under these conditions. The SOA yields as a function of the OA concentration of the
other linear alkanes can be found in Figure S1.

The estimated SOA yield parametrization of the new lumped IVOCs using the five-product VBS are summarized in
Table 3. For the alkane lumped species, we have assumed that the values of $a_i$'s are the same under both high and low NO$_x$
conditions. This is a necessary assumption due to lack of experimental measurements under low NO$_x$ conditions. The
estimated alkane yields increase with size from ALK6 to ALK9 (Figure 8). At OA concentration of 1 μg m$^{-3}$, they range
from 5.6% for ALK6 and 12% for ALK7 to 14% for both ALK8 and ALK9. At an OA concentration of 10 μg m$^{-3}$, the SOA
yields are 9.3%, 34.2%, 42% and 43% respectively. The predicted increase of the estimated yields with molecular size is in
line with experiments that have shown that the SOA yields of cyclic and linear alkanes increase with carbon number (Lim
and Ziemann, 2009; Presto et al., 2010; Tkacik et al., 2012). The estimated values of the organic aerosol yield of ALK8 and
ALK9 are similar because they were based on the n-heptadecane yields, which is the highest linear alkane with experimental
data available. This probably leads to an underestimation of the ALK9 yields.

All the SOA yields of the new alkane species are much higher that the yields of ALK5 which is the biggest lumped
alkane species currently in SAPRC. According to the parametrization of Murphy and Pandis (2009) currently in PMCAMx,





at OA of 1 μg m⁻³, the SOA yield of ALK5 is 0.9%, which is 6 times lower than that of ALK6 and 15 times lower than that of ALK9. At OA concentration of 10 μg m⁻³, the SOA yield of ALK5 is 5%, which is 2 times lower than that of ALK6 and 9 times lower than that of ALK9.

### 3.2.2 PAH Yields

Based on experimental data, the mass-based yields of naphthalene, 1-methylnaphthalene and 2-methylnaphthalene were estimated under both high and low $NO_x$ conditions (Table S3). The yields of these three compounds are shown in Figure 9 as a function of the OA concentration. Under high $NO_x$ conditions, the SOA yields of 1-methylnaphthalene are the highest among the three PAHs. At an OA concentration of 1 μg m⁻³, the estimated SOA yield of naphthalene is 4.2%, of 2-methylnaphthalene it is 4% and of 1-methylnaphthalene 6.4%. At higher OA levels of 10 μg m⁻³, the estimated SOA yields

are respectively 19.3%, 21.2% and 24.5%. Under low $NO_x$ conditions, at OA concentration of 1 μg m⁻³ the SOA yield of 2-methylnaphthalene is 5.7% and for both naphthalene and 1-methylnaphthalene it is 4%. Under the same conditions and at OA of 10 μg m⁻³, 2-methylnaphthalene is estimated to have a yield of 31.4%, 1-methylnaphthalene 23% and naphthalene 22.6%.

       Our parametrization suggests that under low $NO_x$ conditions the yields of 2-methylnaphthalene are higher than

those of naphthalene and 1-methylnaphthalene. Due to the variability of experimental methods used, there has been a lack of consensus among experimental studies about which of the three studied PAHs has the highest and the lowest SOA yields. For example, we estimated that under low $NO_x$ conditions the SOA yields of 1-methylnaphthalene are lower than those of 2-methylnaphthalene. This is in line with observations made by Chan et al. (2009) and Kleindienst et al. (2012), but at the same time it contradicts the results of Shakya and Griffin (2010) and Chen et al. (2016) who propose that the yields of 1-

methylnaphthalene are higher than those of 2-methylnaphthalene. Since all three PAHs are lumped in the same species, PAH1, these experimental discrepancies have little effect on our final yield parametrization.

       For both low and high NOx conditions, the VBS parametrization of the lumped PAHs can be found in Table 3 and the resulted SOA yields as a function of OA are shown in Figure 10. Under high $NO_x$ conditions at organic aerosol concentration of 1 μg m⁻³, the estimated yield of PAH1 is 3.8%% and at organic aerosol concentration 10 μg m⁻³ it is 19%.

For PAH2 the estimated yields are 3.8% and 21% respectively. Under low $NO_x$ conditions for PAH1 and PAH2, at the organic aerosol concentration of 1 μg m⁻³, the estimated yields are 4.6% and 5.7%, at organic aerosol concentrations of 10 μg m⁻³, they are respectively 25% and 31.4%. For both lumped species, the estimated yields under low $NO_x$ conditions are higher than those under high $NO_x$ conditions. This is consistent with the observations which suggest that under high $NO_x$ conditions the PAH's reaction with the OH radical is dominated by the fragmentation route, which leads to higher volatility

products (Chan et al., 2009; Shakya and Griffin, 2010; Kleindienst et al., 2012).





### 3.2.3 Aromatic Yields

The estimated parameters for ARO3 are shown in Table 3. The estimated yields of ARO3 are shown as a function of organic aerosol concentration in Figure 11 together with the yields of ARO2, the lumped species which the mass-based yields of ARO3 were based upon. At the organic aerosol concentration of 1 μg m$^{-3}$, under high NO$_x$ conditions the estimated mass fraction of ARO3 is 8.7% and under low NO$_x$ conditions it is 17.2%. At organic aerosol concentration of 10 μg m$^{-3}$, under high NO$_x$ conditions the estimated aerosol mass fraction of ARO3 is 10.4% and under low NO$_x$ conditions it is 20.6%. Like in the case of the new lumped PAHs, the new aromatic species has higher SOA yields under high NO$_x$ than under low NO$_x$ conditions. This is consistent with experimental observations that have shown that due to the formation of alkoxyl radical, which decompose easier in the atmosphere, SOA formation from aromatics under high NO$_x$ conditions is less effective.

### 4. Conclusions

A new approach for simulating IVOC chemistry and SOA production in CTMs is developed. IVOCs are treated as lumped species similar to the larger VOCs. The new lumping method takes into account both the complex chemistry and the organic aerosol formation potential of the IVOC lumped species. The new lumped scheme that is developed, is consistent with the SAPRC gas-phase chemical mechanism but can also be used in other gas-phase chemistry schemes.

Our estimated IVOC emissions from diesel and gasoline vehicles in Europe are 8 times higher compared to the IVOC emissions previously used assuming that they were equal to 1.5 times the primary OA emissions. Cyclic alkanes have the highest emissions (63% of the total) followed by branched alkanes (15 of the total) and unspeciated aromatic compounds (13% of the total). These compounds are mostly unspeciated and appear usually in the MS/GC measurements as an unresolved complex mixture.

The estimated SOA yields of IVOCs are significantly higher than those of the VOCs currently in the model. Specifically, under high NO$_x$ conditions at organic aerosol concentration of 10 μg m$^{-3}$, the aerosol mass fractions of the new lumped alkanes (ALK6-ALK9) are on average 6.4 times higher than those of ALK5 and the aerosol mass fractions of the new lumped PAHs (PAH1 and PAH2) and ARO3 are on average 9.6 times higher than those of ARO2. Since, the estimates of the SOA yields of the new lumped species are based on data for only nine individual IVOCs, there is significant uncertainty in these values. Smog chamber experiments with the IVOCs that are lumped in the new lumped species are necessary in order to improve the estimates of the yields.

A preliminary application of the new approach to PMCAMx-iv, the results of which will be presented in a subsequent publication, will provide a clearer perspective on which individual IVOCs experimental studies should be focusing on. The sensitivity of the predicted SOA-iv from transportation to our major assumptions will also be examined.



**5. Data availability**

The IVOC emissions inventory and the source code for PMCAMx-iv (v1.0) is available in: https://doi.org/10.5281/zenodo.6515734

**6. Supplementary information**

**7. Author contribution**

SEM and SNP designed the research. SEM developed the lumping scheme, prepared the source specific IVOC emissions over the European domain and designed the SOA parametrization for the new IVOC lumped species. SEM wrote the paper with input from SNP.

**8. Competing interests**

**The authors declare that they have no conflict of interest.9. Funding**

This work has received funding from the European Union's Horizon 2020 research and innovation program under project FORCeS, grant agreement no. 821205 and by the Hellenic Foundation for Research & Innovation (HFRI) under project CHEVOPIN, grant agreement no. 1819.

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



**Table 1:** Characteristics of the new lumped IVOC species.

| Nº Carbons | Compound names | Reaction rate constant with OH $k_{OH}$ (x$10^{-11}$ cm$^3$ molecule$^{-1}$ s$^{-1}$) | Molecular Weight $MW$ (g mol$^{-1}$) | Estimated Saturation Concentration $C^*$ (μg m$^{-3}$) |
|---|---|---|---|---|
| colspan | **ALK6** ($MW$ = 183 g mol$^{-1}$) Alkanes and other non-aromatic compounds that react only with OH ($k_{OH}$: 1.3 - 1.8 x $10^{-11}$ cm$^{-3}$ molecule$^{-1}$ s$^{-1}$) | | | |
| 12 | Dodecane | 1.3 | 170 | $1.8 \times 10^6$ |
| 13 | Tridecane | 1.5 | 184 | $7.3 \times 10^5$ |
| 14 | Tetradecane | 1.7 | 198 | $3.0 \times 10^5$ |
| 14 | 2,6,10-Trimethylundecane | 1.7 | 198 | $1.4 \times 10^6$ |
| 12 | Hexylcyclohexane | 1.8 | 168 | $1.7 \times 10^6$ |
| | Unspeciated b-alkanes B12 | 1.3 | 170 | $1.8 \times 10^6$ |
| | Unspeciated b-alkanes B13 | 1.5 | 184 | $7.3 \times 10^5$ |
| | Unspeciated b-alkanes B14 | 1.7 | 198 | $3.0 \times 10^5$ |
| | Unspeciated cyclic alkanes B12 | 1.3 | 170 | $1.8 \times 10^6$ |
| | Unspeciated cyclic alkanes B13 | 1.5 | 184 | $7.3 \times 10^5$ |
| | Unspeciated cyclic alkanes B14 | 1.7 | 198 | $3.0 \times 10^5$ |
| colspan | **ALK7** ($MW$ = 224 g mol$^{-1}$) Alkanes and other non-aromatic compounds that react only with OH ($k_{OH}$: 1.8 - 2.2 x $10^{-11}$ cm$^{-3}$ molecule$^{-1}$s$^{-1}$) | | | |
| 15 | Pentadecane | 1.8 | 212 | $1.2 \times 10^5$ |
| 16 | Hexadecane | 2.0 | 226 | $4.9 \times 10^4$ |
| 17 | Heptadecane | 2.1 | 241 | $2.0 \times 10^4$ |
| 15 | 2,6,10-Trimethyldodecane | 1.9 | 212 | $5.8 \times 10^5$ |
| 16 | 2,6,10-Trimethyltridecane | 2.1 | 226 | $2.4 \times 10^5$ |
| 13 | Heptylcyclohexane | 1.9 | 182 | $7.1 \times 10^5$ |
| 14 | Octylcyclohexane | 2.1 | 196 | $3.0 \times 10^5$ |
| 15 | Nonylcyclohexane | 2.2 | 210 | $1.2 \times 10^5$ |
| | Unspeciated b-alkanes B15 | 1.8 | 212 | $1.2 \times 10^5$ |
| | Unspeciated b-alkanes B16 | 2.0 | 226 | $4.9 \times 10^4$ |
| | Unspeciated b-alkanes B17 | 2.1 | 241 | $2.0 \times 10^4$ |
| | Unspeciated cyclic alkanes B15 | 1.8 | 212 | $1.2 \times 10^5$ |
| | Unspeciated cyclic alkanes B16 | 2.0 | 226 | $4.9 \times 10^4$ |
| | Unspeciated cyclic alkanes B17 | 2.1 | 241 | $2.0 \times 10^4$ |
| colspan | **ALK8** ($MW$ = 265 g mol$^{-1}$) Alkanes and other non-aromatic compounds that react only with OH ($k_{OH}$: 2.2 - 2.7 x $10^{-11}$ cm$^{-3}$ molecule$^{-1}$ s$^{-1}$) | | | |
| 18 | Octadecane | 2.2 | 255 | $8.1 \times 10^3$ |
| 19 | Nonadecane | 2.4 | 269 | $3.3 \times 10^3$ |





| | | | | |
|---|---|---|---|---|
| 20 | Eicosane | 2.5 | 283 | $1.3 \times 10^3$ |
| 18 | 2,6,10-Trimethylpentadecane | 2.3 | 255 | $4.1 \times 10^4$ |
| 19 | Pristane | 2.4 | 269 | $2.9 \times 10^4$ |
| 20 | Phytane | 2.6 | 283 | $1.2 \times 10^4$ |
| 16 | Decylcyclohexane | 2.3 | 224 | $5.2 \times 10^4$ |
| 17 | Undecylcyclohexane | 2.5 | 239 | $2.2 \times 10^4$ |
| 18 | Dodecylcyclohexane | 2.6 | 253 | $9.1 \times 10^3$ |
| | Unspeciated b-alkanes B18 | 2.2 | 255 | $8.1 \times 10^3$ |
| | Unspeciated b-alkanes B19 | 2.4 | 269 | $3.3 \times 10^3$ |
| | Unspeciated b-alkanes B20 | 2.5 | 283 | $1.3 \times 10^3$ |
| | Unspeciated cyclic alkanes B18 | 2.2 | 255 | $8.1 \times 10^3$ |
| | Unspeciated cyclic alkanes B19 | 2.4 | 269 | $3.3 \times 10^3$ |
| | Unspeciated cyclic alkanes B20 | 2.5 | 283 | $1.3 \times 10^3$ |
| **ALK9** $(MW = 302 \text{ g mol}^{-1})$ Alkanes and other non-aromatic compounds that react only with OH $(k_{OH} > 2.67 \text{ x } 10^{-11} \text{ cm}^{-3} \text{ molecule}^{-1} \text{ s}^{-1})$ | | | | |
| 21 | Heneicosane | 2.7 | 297 | $5.4 \times 10^2$ |
| 22 | Docosane | 2.8 | 311 | $2.1 \times 10^2$ |
| 19 | Tridecylcyclohexane | 2.8 | 267 | $3.8 \times 10^3$ |
| 20 | Tetradecylcyclohexane | 2.9 | 281 | $1.6 \times 10^3$ |
| 21 | Pentadecylcyclohexane | 3.0 | 295 | $6.4 \times 10^2$ |
| 22 | Hexadecylcyclohexane | 3.2 | 309 | $2.6 \times 10^2$ |
| 23 | Heptadecylcyclohexane | 3.3 | 323 | $1.1 \times 10^2$ |
| | Unspeciated b-alkanes B21 | 2.7 | 297 | $5.4 \times 10^2$ |
| | Unspeciated b-alkanes B22 | 2.8 | 311 | $2.1 \times 10^2$ |
| | Unspeciated cyclic alkanes B21 | 2.7 | 297 | $5.4 \times 10^2$ |
| | Unspeciated cyclic alkanes B22 | 2.8 | 311 | $2.1 \times 10^2$ |
| **ARO3** $(MW = 188 \text{ g mol}^{-1})$ Larger aromatics | | | | |
| 11 | Pentylbenzene | 1.0 | 148 | $2.4 \times 10^6$ |
| 12 | Hexylbenzene | 1.2 | 162 | $1.0 \times 10^6$ |
| 13 | Heptylbenzene | 1.3 | 176 | $4.1 \times 10^5$ |
| 14 | Octylbenzene | 1.4 | 190 | $1.7 \times 10^5$ |
| 15 | Nonylbenzene | 1.6 | 204 | $6.9 \times 10^4$ |
| 16 | Decylbenzene | 1.7 | 218 | $2.8 \times 10^4$ |
| 17 | Undecylbenzene | 1.9 | 232 | $1.2 \times 10^4$ |
| 18 | Dodecylbenzene | 2.0 | 246 | $4.7 \times 10^3$ |
| 19 | Tridecylbenzene | 2.1 | 261 | $1.9 \times 10^3$ |
| 20 | Tetradecylbenzene | 2.3 | 275 | $7.7 \times 10^2$ |
| 22 | Pentadecylbenzene | 2.4 | 289 | $3.1 \times 10^2$ |
| **PAH1** $(MW = 137 \text{ g mol}^{-1})$ Polycyclic aromatic hydrocarbons with $k_{OH} < 7 \text{ x } 10^{-11} \text{ cm}^{-3} \text{ molecule}^{-1} \text{ s}^{-1}$ | | | | |
| 10 | Naphthalene | 1.6 | 128 | $1.8 \times 10^8$ |
| 11 | 2-methylnaphthalene | 4.9 | 142 | $7.1 \times 10^7$ |
| 11 | 1-methylnaphthalene | 4.1 | 142 | $7.1 \times 10^7$ |





| 12 | C2-naphthalene | 6.0 | 156 | $3.5 \times 10^7$ |
|---|---|---|---|---|
| 13 | Fluorene | 1.6 | 166 | $1.3 \times 10^6$ |
| 14 | Phenanthrene | 3.2 | 178 | $1.1 \times 10^8$ |
| 15 | C1-Phenanthrene | 5.9 | 192 | $4.7 \times 10^7$ |
| 16 | Fluoranthene | 3.3 | 202 | $3.7 \times 10^8$ |
| 16 | Pyrene | 5.6 | 202 | $3.0 \times 10^8$ |
| | Unspeciated aromatic compounds B12 | 1.6 | 128 | $1.8 \times 10^8$ |
| | Unspeciated aromatic compounds B13 | 4.9 | 142 | $7.1 \times 10^7$ |
| | Unspeciated aromatic compounds B14 | 6.0 | 156 | $3.5 \times 10^7$ |
| **PAH2** ($MW = 175$ g mol$^{-1}$) Polycyclic aromatic hydrocarbons with $k_{OH} > 7 \times 10^{-11}$ cm$^{-3}$ molecule$^{-1}$ s$^{-1}$ | | | | |
| 12 | Acenaphthylene | 12.4 | 152 | $1.3 \times 10^7$ |
| 12 | Acenaphthene | 8.0 | 154 | $1.4 \times 10^7$ |
| 13 | C3-naphthalene | 8.0 | 170 | $1.1 \times 10^7$ |
| 13 | C4-naphthalene | 8.0 | 184 | $4.4 \times 10^6$ |
| 14 | C1-Fluorene | 8.0 | 180 | $5.1 \times 10^5$ |
| 14 | Anthracene | 17.8 | 178 | $1.1 \times 10^8$ |
| 16 | C2-Phenanthrene/anthracene | 8.0 | 192 | $4.7 \times 10^7$ |
| 17 | C1-Fluoranthene/pyrene | 13.1 | 202 | $3.0 \times 10^8$ |
| | Unspeciated aromatic compounds B15 | 8.0 | 170 | $1.1 \times 10^7$ |
| | Unspeciated aromatic compounds B16 | 8.0 | 184 | $4.4 \times 10^6$ |




**Table 2:** Experimental smog chamber studies utilized in this work to estimate the mass-based stoichiometric yields of the individual IVOCs.

| Compounds studied | Experimental conditions | Reference |
|---|---|---|
| n-Dodecane, n-Tridecane, n-Tetradecane, n-Pentadecane, n-Hexadecane, n-Heptadecane | T = 298 K, exp. With RH<1% and exp. with RH>15%, High NO$_x$ conditions, High C$_{OA}$ loadings | Lim and Ziemann (2009) |
| n-Dodecane, n-Pentadecane, n-Heptadecane | T = 295 K, RH < 20 %, High NO$_x$ conditions | Presto et al. (2010) |
| n-Dodecane, n-Tridecane, n-Tetradecane | T=298 K, RH varied, High NO$_x$ conditions, No wall corrections | Docherty et al. (2021) |
| Naphthalene, 1-Methylnaphthalene, 2-Methylnaphthalene, | T=299 K, RH from 5% to 8%, High and Low NO$_x$ conditions | Chan et al. (2009) |
| Naphthalene, 1-Methylnaphthalene, 2-Methylnaphthalene | T=21-25 ºC, RH<5%, High and Low NO$_x$ conditions | Shakya et al. (2010) |
| Naphthalene, 1-Methylnaphthalene, 2-Methylnaphthalene | T=298 K, RH < 3% High and Low NO$_x$ conditions | Kleindienst et al. (2012) |
| Naphthalene, 1-Methylnaphthalene, 2-Methylnaphthalene | Dry conditions, High and Low NO$_x$ conditions | Chen et al. (2016) |






**Table 3:** Aerosol mass-based yields for the new lumped species in the IVOC range using a five-product basis set
with saturation concentrations of 0.1, 1, 10, 100 and $10^3$ µg m$^{-3}$ at 298 K.

| Lumped Species | Aerosol mass-based yields | | | | |
|---|---|---|---|---|---|
| | 0.1 µg m$^{-3}$ | 1 µg m$^{-3}$ | 10 µg m$^{-3}$ | 100 µg m$^{-3}$ | $10^3$ µg m$^{-3}$ |
| **High-NO$_x$ Parametrization** | | | | | |
| **nALK6** | 0.038 | 0.035 | 0.03 | 0.062 | 0.309 |
| **ALK7** | 0.025 | 0.117 | 0.401 | 0.103 | 0.052 |
| **ALK8** | 0.074 | 0.029 | 0.622 | 0.148 | 0 |
| **ALK9** | 0.077 | 0.024 | 0.629 | 0.151 | 0 |
| **ARO3** | 0 | 0.001 | 0.156 | 0.24 | 0.348 |
| **PAH1** | 0 | 0.011 | 0.346 | 0.038 | 0.083 |
| **PAH2** | 0 | 0 | 0.407 | 0.077 | 0.163 |
| **Low-NO$_x$ Parametrization** | | | | | |
| **ALK6** | 0.038 | 0.035 | 0.03 | 0.062 | 0.309 |
| **ALK7** | 0.025 | 0.117 | 0.401 | 0.103 | 0.052 |
| **ALK8** | 0.074 | 0.029 | 0.622 | 0.148 | 0 |
| **ALK9** | 0.077 | 0.024 | 0.629 | 0.151 | 0 |
| **ARO3** | 0 | 0.06 | 0.24 | 0.3 | 0.42 |
| **PAH1** | 0 | 0.003 | 0.484 | 0 | 0.032 |
| **PAH2** | 0 | 0 | 0.627 | 0 | 0.074 |

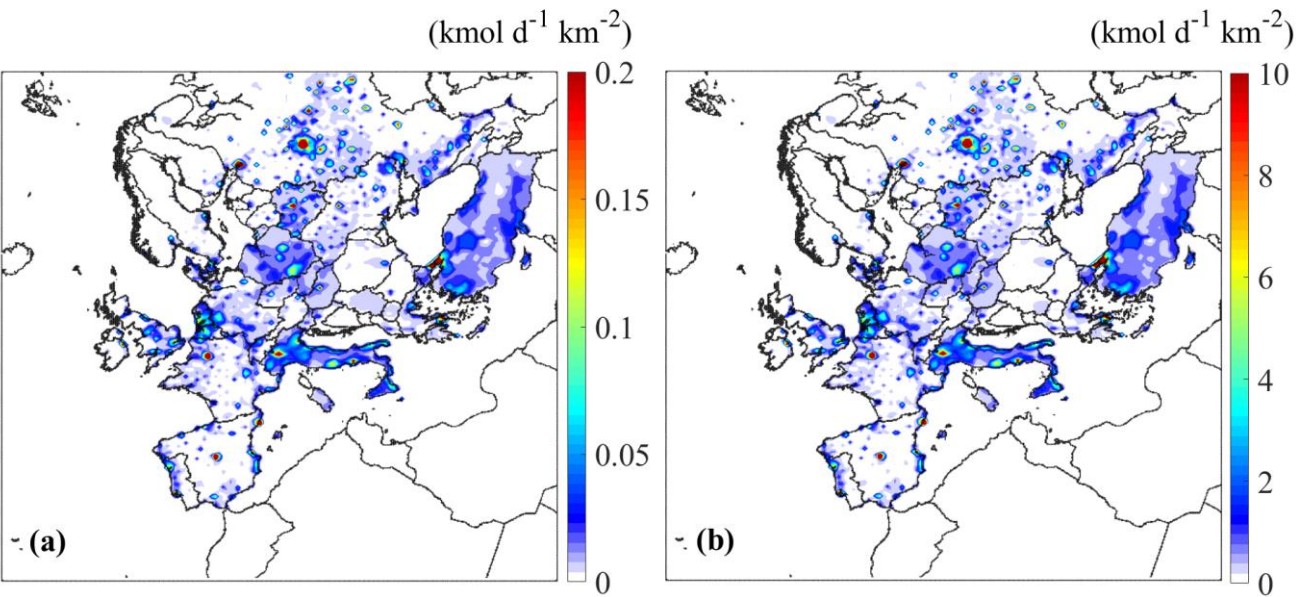

**Figure 1:** Estimated spatial distribution of the averaged emissions from on-road vehicles of: (a) n-Dodecane and (b) ALK6 for May 2008.



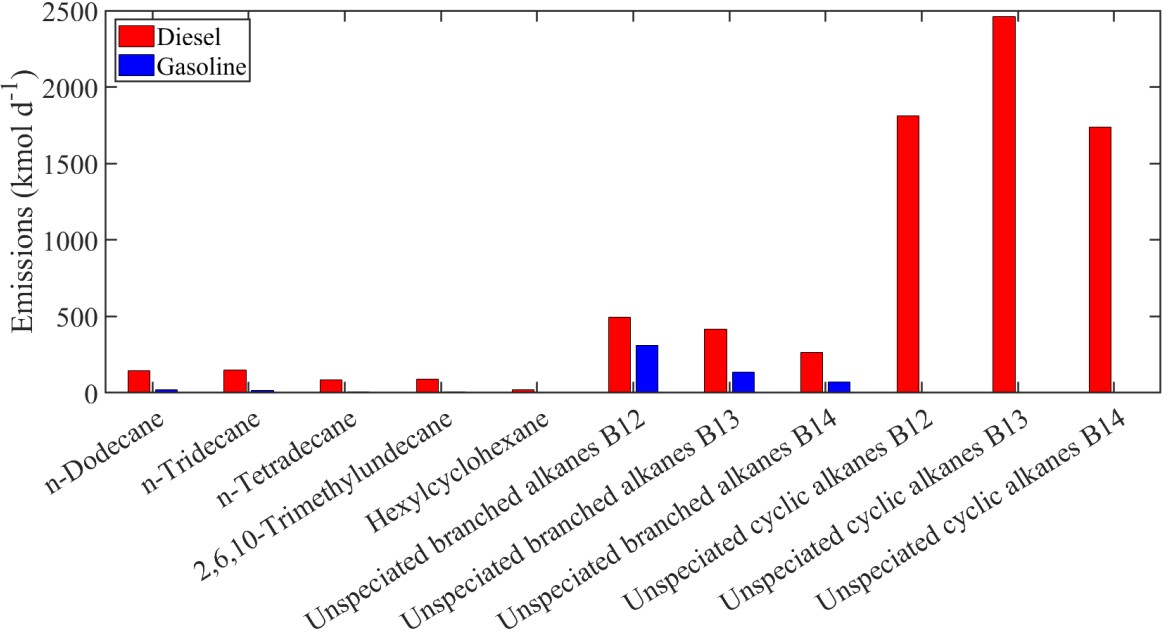

**Figure 2:** Estimated total gasoline and diesel emissions of the new individual compounds lumped into ALK6 for Europe


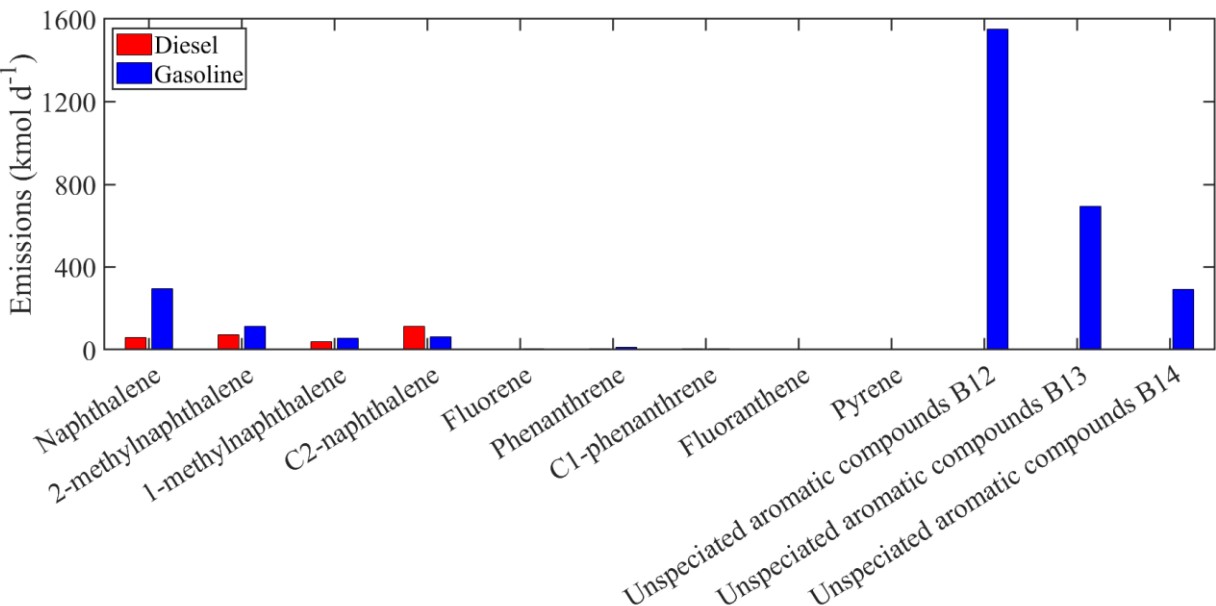

**Figure 3:** Estimated total gasoline and diesel emissions of the new individual compounds lumped into PAH1 for Europe.


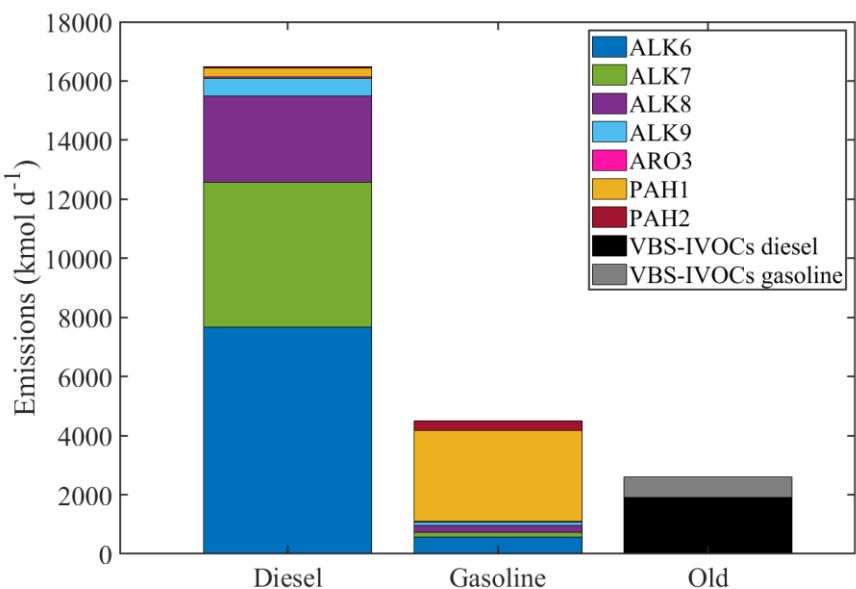

**Figure 4:** Total emissions from diesel and gasoline vehicles over Europe for May 2008 calculated using the new lumped species approach and using the old VBS approach (signified with grey and black).






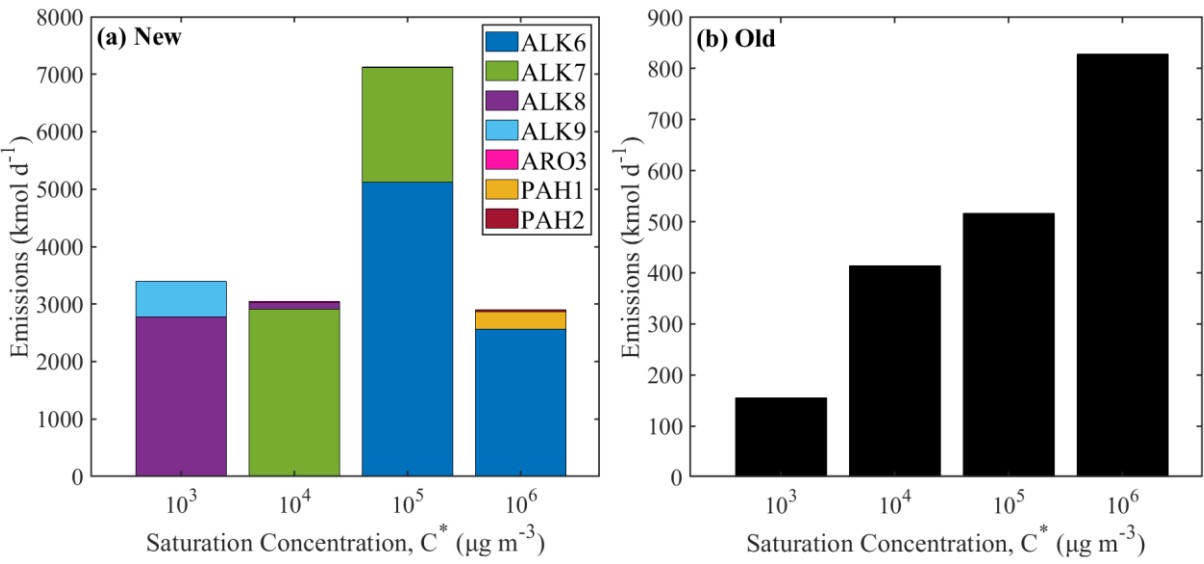

**Figure 5:** The volatility distribution of the IVOCs emissions from on-road diesel vehicles (a) using the new lumped species approach and (b) using the old VBS approach. Different axes are used for the two distributions.

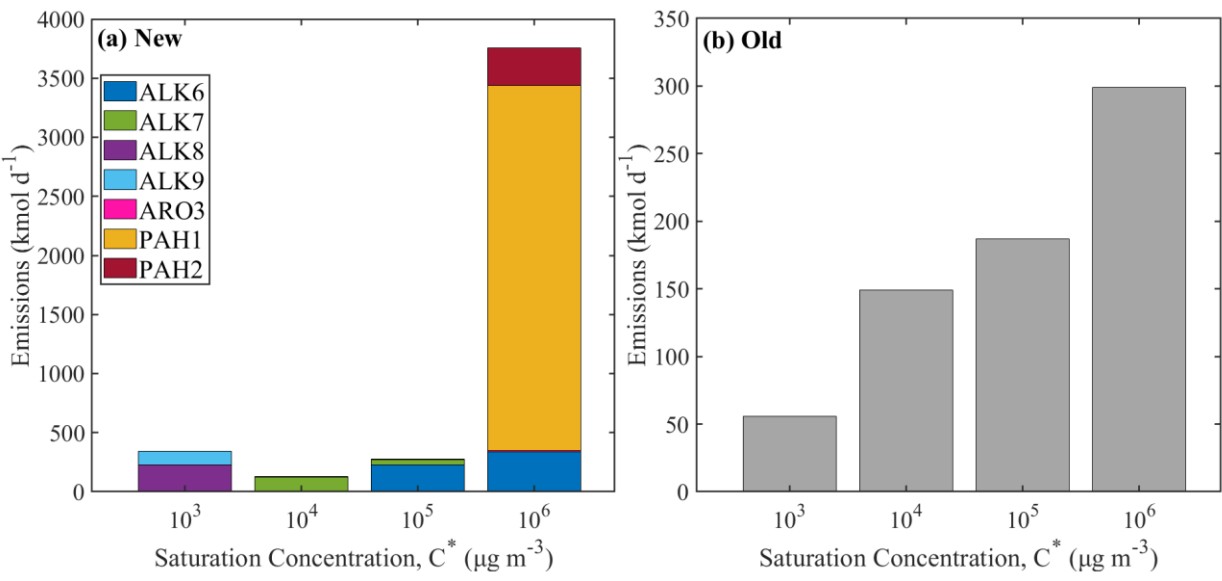


**Figure 6:** The volatility distribution of the IVOC emissions from on-road gasoline vehicles (a) using the new lumped species approach and (b) using the old VBS approach. Different axes are used for the two distributions.



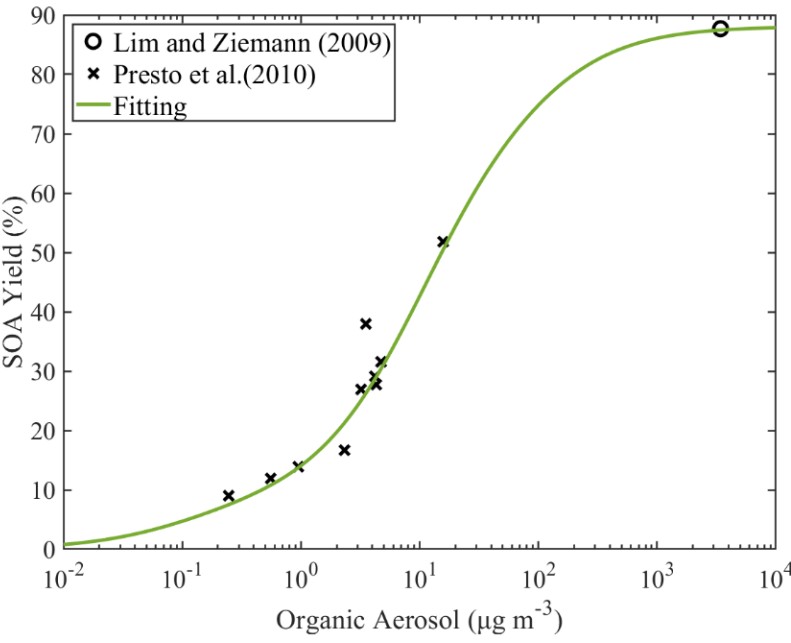

**Figure 7:** Estimated SOA yields of n-Heptadecane under high NOx conditions. The curve is generated using the estimated ai (0.0771, 0.024, 0.6291, 0.1506 and 0). The data from the studies of Lim and Ziemann (2009) and Presto et al. (2010) that were used by the simplified algorithm are also shown.

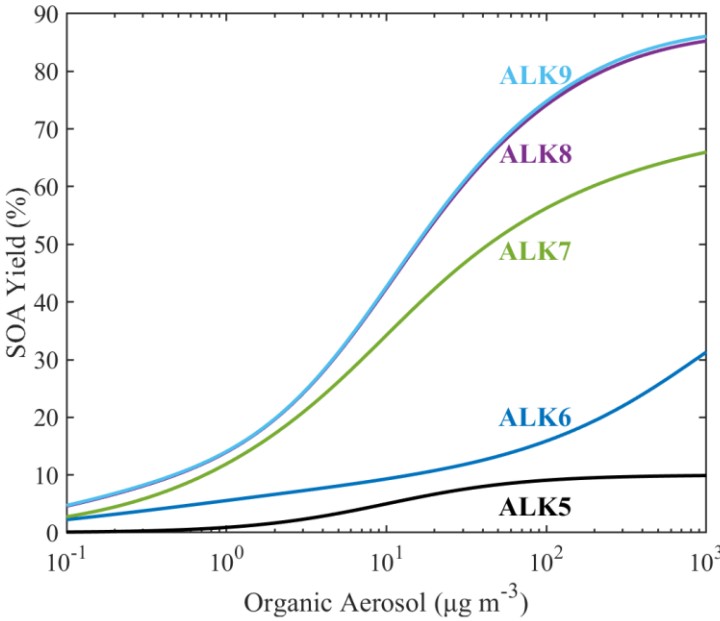

**Figure 8:** Estimated SOA yields of the new lumped alkane species under high NOx conditions. The curves are
generated using the SOA yield parametrization of Table 3.



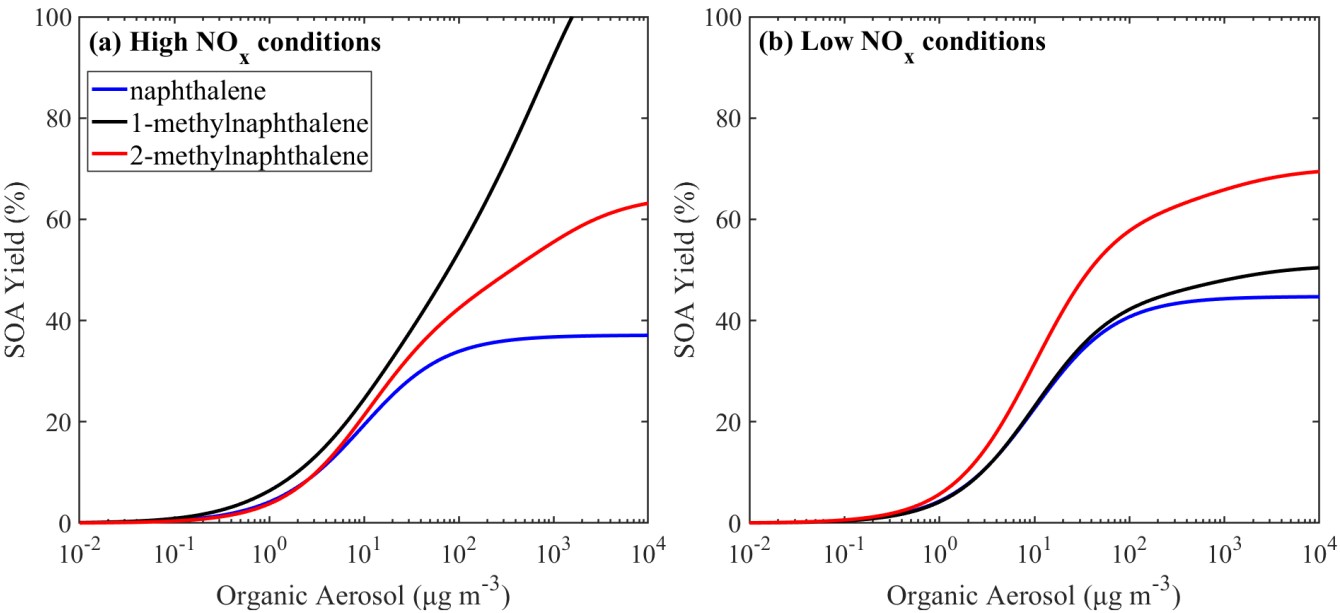

**Figure 9:** Estimated SOA yields of naphthalene, 1-methylnaphthalene and 2-methylnaphthalene under (a) high NOx and (b) low NO$_x$ conditions. The curves are generated using the estimated parameters of Table S2.

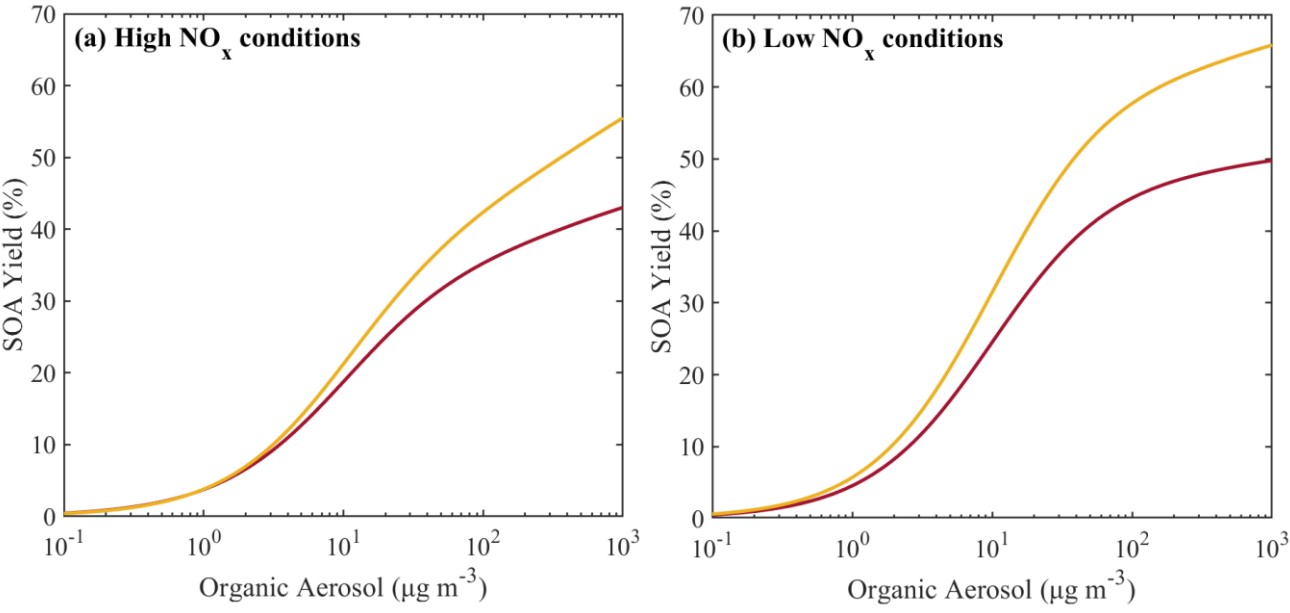


**Figure 10:** Estimated aerosol mass fractions of the new lumped PAH species under (a) high NO$_x$ and (b) low NO$_x$ conditions. The yield curves are generated using the mass-based yields of Table 3.



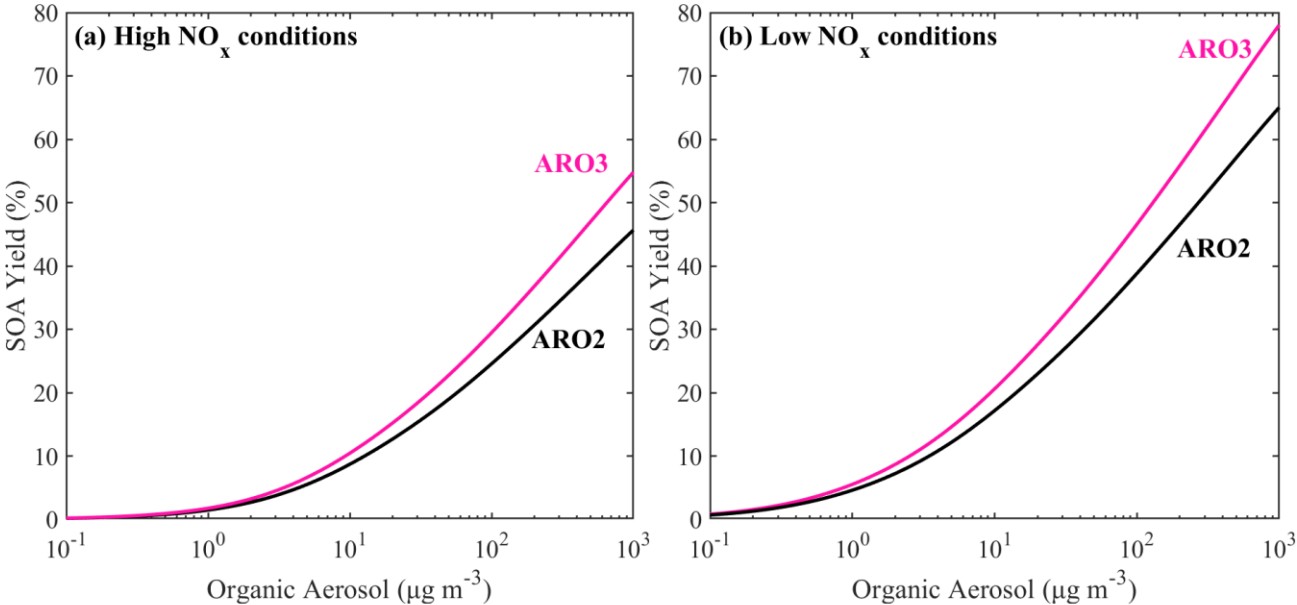

**Figure 11:** Estimated aerosol yields for the new aromatic species (ARO3) and the existing ARO2 species under
(a) high NO$_x$ and (b) low NO$_x$ conditions.