# Peer review of "A lumped species approach for the simulation of secondary organic aerosol production from intermediate volatility organic compounds (IVOCs): Application to road transport in PMCAMx-iv (v1.0)"

_Geoscientific Model Development, 2022_

## Author Response (AR1)

**Responses to the Comments of Referee #1**

**General comments**

**1.** This study presents a new model to represent the IVOC emissions and SOA formation from on-road gasoline and diesel vehicles in Europe. A SAPRC-based model using 7 new surrogate species was developed based on experimental emissions and smog chamber data. This study shows that IVOC emissions have previously been underestimated by a factor of 8, and the new model surrogates have higher SOA yields than existing VOC model surrogates. The paper does a good job of describing how the new model was developed, but I have a few general questions about the Introduction and Conclusion, and recommendations for how to reorganize some of the information.

We appreciate the positive assessment of our work by the referee. Following the recommendations of the reviewer, we have revised the paper (including the Introduction, the Conclusion and the SI) to provide additional information about the new model, its parameters and required emissions. These changes are described below (in black) following each comment of the reviewer (in blue).

**Specific Comments**

**2.** In this paper you focus specifically on IVOCs. Can you please add a statement in the Introduction stating why you do not investigate SVOCs, LVOCs, or ELVOCs? In Section 2.2 (line 104), you state that the existing PMCAMx VBS model includes volatility bins reaching to 0.01 µg m$^{-3}$, so it would be good to specifically (briefly) address those compounds as well. I believe lower-volatility species simply were not included in the Zhao work that your model is parameterized on?

This is a good point. In our previous work during the last 15 years, the VOCs were simulated as independent or lumped species in the corresponding gas-phase chemistry mechanism (SAPRC in our case) and their chemistry was described in some detail. The remaining organics were simulated using the Volatility Basis Set framework following only their volatility distribution and their chemistry was oversimplified. In this work, we take advantage of the developments during the last decade to perform the next step and to add the IVOCs to the species that are simulated with some detail. Unfortunately, it is still too early to simulate the SVOCs and LVOCs explicitly (as lumped species) due to the lack of the necessary information. Hopefully, this will become possible in the near future. We have followed the advice of the reviewer and added this information in the Introduction.

**3.** Line 26-28: Emitted/primary IVOCs also include oxygenated compounds, although these tend to not be emitted from vehicles.

Indeed, oxidized compounds in the IVOC range, such as phenols and substituted phenols, have been identified in the emissions of biomass burning. As the reviewer points out oxidized IVOCs have not been detected in neither the emissions of on-road vehicles nor in the emissions of other diesel or gasoline powered engines. Additional IVOC species will be needed as our approach is extended to other sources, but the framework is flexible enough to accommodate them easily. This point is now discussed in the corresponding section of the revised paper.

**4.** Line 44: "due to their size and low volatility"

We have made the suggested change.

**5.** Lines 63-79: Please give more description why the models described here, which use the same experimental dataset, are insufficient. For example, the model from Lu et al., 2020 already separates the IVOC emissions by C*, kOH, MW, SOA yield, and structure (alkane vs. aromatic). Your model further separates IVOCs into more specific structural categories, which is a difference worth explicitly mentioning in this paragraph and the Conclusion.

We have followed the suggestion of the reviewer and revised the introduction (lines 63-79) to include a more detailed description of the model developed by Lu et al. (2020). We have also included a description of the differences between our model and the model of Lu et al. (2020) in the Conclusions section, as indicated by the reviewer. The model of Lu et al. (2020), like our model, treats IVOCs as lumped species and it, also like our model, utilizes the work of Zhao et al. (2015; 2016). However, in our model we have chosen to expand on the idea of Lu et al. (2020), by separating the 79 compounds identified by Zhao et al. (2015; 2016) into lumped species based on more specific categories depending on the structure of the molecules and their chemical reactivity. These new lumped species are easily added to the SAPRC gas-phase chemistry mechanism, because they are consistent with its overall structure and philosophy. It should also be noted that the yields and the reaction rate constants with the hydroxyl radical of the new lumped species differ in the two approaches models. We should also point out that we do not consider the Lu et al. (2020) approach to be any means insufficient.

**6.** Table 1 is very detailed and may be more appropriate for the SI. Table 1 could instead just give the properties of the 7 new surrogates used in the model (C*, kOH, MW, SOA yields/$a_i$).

Table 1 presents detailed information about the individual compounds lumped into each one of the new IVOC species. Despite its length, we would prefer to keep it in the main paper because it will help link the individual species with the lumped compounds in the proposed mechanism. Information about the estimated SOA yields is provided in Table 3.

**7.** Reference Table 1 (or Table S1, if you move it to the SI) in the second paragraph of Section 2.3 to introduce your descriptions beginning in line 122.

This is a good suggestion. We have made the appropriate changes and now reference Table 1 before introducing a detailed description of the individual compounds lumped in the new IVOC species in the second paragraph of Section 2.3.

**8.** Line 175-177: This is somewhat vague. State explicitly here what the "more volatile products of the reactions" are or state what the "larger lumped VOCs that are already present in the SAPRC mechanism" are. This may warrant an extra table in the SI matching up existing surrogate products with the new surrogate products, or reference the code provided in the Data Availability section.

We now clarify at this point that the "more volatile products" are the gas-phase oxidation products of the various IVOCs such as peroxy radicals, ketones, aldehydes, glyoxal and others, whereas the "larger lumped VOCs" refer to ALK5 and ARO2. Moreover, as suggested by the reviewer, we have introduced a new table in the SI that provides the composition of the already existing nine anthropogenic and two biogenic lumped species in the SAPRC mechanism.

**9.** Lines 182 and 194: Are the OCGi species the same as the 5 products that you reference in lines 173-174?

Yes, the term $OCG_i$ refers to the five lower volatility products that can partition to the aerosol phase forming SOA. All seven oxidation reactions of the new lumped species contribute to the formation to the same five secondary products in the model. This is now clarified in the revised paper.

**10.** Lines 189 and 198: The reactions of ALK7-ALK9 and PAH1-PAH2 should be given somewhere (an in-text or SI table, or referenced to the code provided in the Data Availability section).

The reactions of the other lumped species are included in a new table in the SI in the revised manuscript.

**11.** I recommend reorganizing Sections 2.4.1-2.4.3 and 2.5. Presenting the information more chronologically could be helpful. Reference Section 2.5 in the two sentences in lines 206-209.

We have changed the order of presentation of the corresponding sections and also reorganized the material in them. We first describe the emissions, then the measurement data, the algorithm used for the fitting and finally the resulting SOA-iv yields. We do connect now the material in the yields section with that of the emissions section.

**12.** Line 300 and Figure 1: Are the spatial and temporal distribution of the IVOC emissions determined by the GEMS inventory given in line 292? The spatial and temporal distribution warrant more description since you present the maps in Figure 1.

Indeed, the spatial and temporal distributions of the IVOC emissions are based on the GEMS inventory. This is necessary because the new IVOC emissions are calculated based on the total VOC emissions from on-road diesel and gasoline vehicles. We have made the necessary changes in Section 3.1 to describe in more detail the temporal and spatial distributions of n-dodecane emissions from on-road diesel and gasoline vehicles over Europe. We also include now in the revised SI a figure with the temporal evolution of the n-dodecane emissions over Paris for a month.

**13.** Consider adding figures to the SI for all new surrogates which match the information given in Figures 2 and 3.

We have followed the suggestion of the reviewer and now include five new figures in the SI that depict the estimated total gasoline and diesel emissions of the compounds lumped into ALK7-ALK9, ARO3 and PAH2.

**14.** Line 353: "Estimated based on experimental data and the fitting algorithm"

We have revised the corresponding sentence.

**15.** Lines 355-356: These yields do not match the values given in Table S3. Are these values from the fitted line?

We now clarify that the yields discussed here are the total SOA yields at a specific concentration and not the yields of the individual VBS species. So indeed, these values are from the fitted line (total yield) and not from Table S3 (yields of individual VBS species).

**16.** Lines 408-409: Add reference to the last statement.

We have added a reference about the differences of the SOA yields of aromatic compounds under high and low $NO_x$ conditions.

**17.** Lines 413-414: A benefit of your model is that it matches the same surrogate+reaction scheme of the existing SAPRC model, so it could be easily integrated into existing SAPRC models rather than integrating an entirely new VBS or other model. I think this benefit should be more explicitly stated in the Conclusion.

We have followed the suggestion of the reviewer and we now highlight this benefit in the revised Conclusions section.

**18.** Can you provide a quantitative estimate of how much this model could increase predicted SOA mass in Europe? In lines 427-230 you state that a subsequent study will apply the model, but using the results in this paper you can predict a bulk percent increase of SOA mass over Europe.

The increase in predicted SOA obviously depends on the simulated conditions. Based on a simple calculation taking into account the new yields and emissions we estimate that the increase would be of the order of 50% for the IVOCs emitted from on-road diesel and gasoline vehicles compared to the default VBS approach currently used in PMCAMx. However, because this is a rough estimate with numerous assumptions, we would prefer to present it in detail in the forthcoming publication.

**Technical Corrections**

**19.** All: Some of the in-text citations use et al. and others use et al
We have corrected all the in-text citations to the appropriate "et al.".

**20.** Lines 89-90: Replace "on" with "to" to avoid the repetition of "on on-road": "In this work, the proposed IVOC scheme is applied to on-road transportation and more specifically to IVOCs emitted by diesel and gasoline vehicles following the studies of Zhao et al. (2015; 2016)."
We have replaced the preposition from "on" to "to".

**21.** Line 93: Name the version of SAPRC, e.g. SAPRC07 or SAPRC99.
In the current version of PMCAMx, we are utilizing a modified version of SAPRC99. This is now mentioned in the text.

**22.** Line 166: "below"
We have corrected the typo.

**23.** Line 201: Be consistent with the tense used. "includes" should be "included".
Since the whole paragraph is written in the present tense, we have corrected the tense in the following sentence from "was used" to "is used".

**24.** Line 296: Define the EUCAARI acronym and give a reference.
The definition of the acronym EUCAARI (European Aerosol Cloud Climate and Air Quality Interactions project) has been added.

**25.** Figure 4: Rename x-axis label "Old" to "Old VBS" for clarity and consistency with the text.
We have renamed the x-axis label of the figure to improve consistency.

**26.** Figure 7: ai (subscript)
We have corrected the typo.

**27.** Figure 10: Add legends.
We have added the appropriate legends to the figure.

**28.** Lines 415-416: "…compared to the IVOC emissions previously used, which assumed that they were equal to…"
We have corrected the syntax error appearing in lines 415-416.

**29.** Line 417: "15%"
We have corrected the typo.

**30.** Line 418: You use MS/GC without defining the acronym, but in line 32 you use gas-chromatograms.
We have defined the MS/GC (mass-spectrometry / gas-chromatography) acronym.

**31.** In the Zenodo link, correct "were" to "where" in the description and correct the title.
The appropriate changes were made to the Zenodo link.

**Responses to the Comments of Referee #2**

**1.** The paper by Manavi and Pandis presents an interesting framework for a lumped oxidation scheme of IVOC (intermediate volatility organic compounds) compounds and subsequent SOA formation. The scheme relies on a review of recent laboratory studies giving SOA yields for oxidation of parent IVOCs. Such a scheme could be implemented into CTMs for better predicting SOA build-up from IVOC emissions. It shows that IVOC emissions lead to much higher SOA yields as from the highest class of alkane and aromatic compounds in the SPARC chemical mechanism. IVOC emissions from the road sector are also evaluated over Europe. The paper should be of interest for the GMD readership and is recommended for publication after several issues have been addressed.

We do appreciate the positive assessment of out manuscript. We have made several changes to the revised text (including the introduction, conclusions and methods sections) to improve it and to make the description of our approach clearer. These changes are described below (in black) following each comment of the reviewer (in blue).

**General comments**

**2.** The paper is somewhat in between a presentation of a general IVOC oxidation framework and an implementation into PMCAMx, for the case of traffic emissions. As such a case study is reserved for a follow-up paper (this seems acceptable), I suggest to present the development as a general one, being applicable to CTM's in general. Then there is no need to explore the case study and May 2008 European IVOC emissions. For instance, the emission section 2.5 is clearly engaged in explaining how emission data for the European wide case study are built (however lacking details, see my remark below). I would take a step back, and focus on what is provided by the new IVOC module, how these data can be specific for US, as you already say, and what data is needed to couple this with classical emission data.

We have revised the presentation of our work, where appropriate, so that it refers in general to CTMs and not only to PMCAMx. We do prefer to keep the presentation of the emissions for Europe, both as an illustration of the approach and also as a way to show the magnitude of changes that this new framework brings to the CTM inputs and therefore indirectly show its potential importance. We have also added a short discussion, as suggested by the reviewer, about the data needed for the coupling of our framework with classical emission data.

**3.** My other concern is the benchmark against which the new scheme is evaluated. The proposed benchmark here for yields are the SPARC ALK5 and ARO2 classes for heaviest or heavier alkane and aromatic VOC's. Doesn't this approach mean that you compare to the new model formulation to a model without implementation of IVOC related SOA formation. I would have expected an evaluation against the Robinson et al. (2007) VBS IVOC scheme, which albeit simple, has been probably implemented in many CTMs using the VBS approach. Sure, reference to this scheme is made a many places in the paper, but a comprehensive and quantitative evaluation of differences is not made, and would be an interesting add-up of the study. This could be done in a 0D mode, for example, but not necessarily, starting from average European May 2008 conditions. This would allow discussing how the interplay between higher IVOC emissions, lower kOH rate constants and higher SOA yields in the new scheme presented here affects SOA yields with respect to the former Robinson scheme and also to a scheme without IVOC emissions.

We do agree with the main point of the reviewer, that the proposed scheme should be compared against the VBS scheme of Robinson et al. (2007). This is actually done for the emissions (see for example Figures 4-6) in the current paper. The comparison for the yields is challenging

because the Robinson et al. (2007) scheme produces SOA through a series of aging reactions shifting the volatility of each product generation by one order of magnitude for each generation of reactions. While this original VBS scheme can form significant amounts of SOA-iv, it does it rather slowly and therefore the SOA-iv levels predicted are also affected by the considerable dilution that takes place in similar timescales as the reactions. For this reason, the comparison of the results of the two schemes should be performed in a 3D model, so that one can avoid potential pitfalls due to the oversimplification of important processes. This comparison is performed in detail in the forthcoming manuscript.

We have made the necessary changes to the revised manuscript to clarify that the original VBS is an appropriate benchmark for the new approach. We also clarify that we do not use the ALK5 and ARO2 species as benchmarks. We compare the corresponding yields mainly to better connect the new IVOCs with the existing VOCs in the model. These comparisons also allow us emphasize the fact that although the IVOCs have lower emissions than VOCs, their higher yields suggest that they can be important SOA precursors.

**4.** Another fundamental difference between the ancient Robinson and the new scheme is that the Robinson scheme only moves mass to next lower volatility class while in the new scheme products can have a large range of volatility. A question: once these IVOC oxidation products formed, do they still further age and subsequently pass to lower volatility classes, as in the initial Robinson scheme?

This a good point that needs clarification. In the proposed scheme, the oxidation products of IVOCs do not undergo any further aging once they are formed. However, because we recognize that such reactions may be important, these multigenerational aging reactions have been included in the PMCAMx-iv (v1.0) code but with their rate constants set equal to zero. This is now explained in the revised paper.

**5.** Following the philosophy of a new module that could be implemented in diverse CTM models, it would be interesting add a short section indicating the needed model structure and input.

Following the recommendation of the reviewer, the revised manuscript includes a short description of the model requirements needed to implement our new lumped species approach.

**Specific comments**

**6.** The paper's title is «A lumped species approach for the simulation of secondary organic aerosol production from intermediate volatility organic compounds (IVOCs): Application to road transport in PMCAMx-iv (v1.0)». From this I would expect that the new scheme is run and evaluated at least for a case study, but the paper is restricted to model formulation (which is OK). The scheme can also be implemented to other CTM's quite directly as long as they use the SPARC chemical mechanism. I would suggest to make appear this more general aspect in the paper's title.

Following the suggestion of the reviewer, we have revised the presentation of our work, where appropriate, so that it refers in general to CTMs and not only to PMCAMx. We understand the point regarding the title of the paper, but given the editorial policy of GMD regarding the model title and version number, the current title is the best solution. We have revised the Conclusions section to emphasize the benefit of integrating our proposed mechanism to models that utilize the SAPRC gas-phase chemical mechanism.

**7.** Page 6, line 178: The volatile products of the reactions of the four new lumped alkane species are assumed as a zeroth approximation to be the same as the ones produced by the reaction of

ALK5. Even if this may underestimate mass of gaseous products, still this allows to stick as close as possible to existing gas phase chemistry (still some differences due to different OH reactivities). May be worthwhile to say.

This is a valid point. Indeed, in our approach, the representation of the gas-phase chemistry of IVOCs is significantly improved. We have followed the suggestion of the reviewer and revised the manuscript accordingly.

**8.** Page 8, line 215 : the following objective function Q:

$$Q = \sum_i [Y_{i,meas} - Y_{i,pred}(a_i, \Delta H)]^2$$

where $Y_{i,meas}$ are the measured aerosol SOA yields and $Y_{i,pred}$ is the corresponding predicted yield for the choices of the parameters, using the VBS framework. The objective function $Q$ is minimized by using the *fmincon* MATLAB function (MathWorks, 2020). By minimizing the objective function, the optimal $\Delta H$ and $a_i$'s are determined for the chosen $C_i^*$ basis set. »

How accurate is this method? The method ideally requires that laboratory results cover the range of OA concentrations from 0.1 µg/m$^3$ to 1000 µg/m$^3$. For some of the compounds in Figure S1, laboratory studies do not cover atmospheric relevant conditions with low enough OA concentrations. Such problems also might have appeared for former studies in the VBS framework, but still the authors should please comment to this question and put some sentences about the limitations of the method.

We agree with the reviewer that the accuracy of the corresponding yields for atmospheric conditions is determined to a large extent by the availability of the necessary experimental data. The chemical detail of our approach can help here as it can point out to the most important IVOCs for SOA production and thus guide future experimental investigations. This point is discussed briefly in the revised paper.

**9.** Page 8, line 230: « For the individual compounds lumped in ALK7, ALK8, PAH2 and ARO3 there were no experimental data. » I think it should be ALK8 and ALK9 instead of ALK7 and ALK8. A few lines above you say that there are data for ALK7.

We have corrected the typo in this sentence.

**10.** Page 9, line 249:
« For example, the mass-based yields of 2,6,10-trimethyltridecane are assumed to be the same as these of n-tridecane. This provides a lower bound for our estimations, as it has been suggested that the SOA yields decrease as the number of branching methyl groups increase … » Shouldn't it be a higher bound, following your argumentation?

This sentence is indeed confusing. We have deleted the last part of the sentence: "as it has been suggested …increase.". The statement is correct, but it refers to the total number of carbon atoms.

**11.** Page 9, section 2.5: In this section, authors describe IVOC emission factors for specific US conditions, which may be a necessary assumption. In Figure 1, they give spatialised emissions distributions for some compounds and for a given month. Authors should indicate the sources of data needed for such estimations (estimations of total VOC and traffic emissions, vehicle fleet partition for the different classes in Table S1, etc.). The simple reference to the former GEMS project is not sufficient. From table S1, it appears that finally only gasoline and diesel cars are distinguished neglecting differences between passenger and light, medium and heavy duty cars. It seems that bulk European fraction of diesel cars is used, but these fractions are different from country to country. May be at the end giving spatialised emission estimates goes to far given the limited data for this study, and this can be left for the follow-up 3D study.

We have made the appropriate changes to the manuscript and the SI to provide a more detailed description of the method that we utilized to estimate the new IVOC emissions over the European domain. The method does rely on the existence of a spatially distributed inventory for transportation (the GEMS inventory in our case), because it follows its spatial and temporal patterns. Additional information used for the derivation of these spatial and temporal dependencies in GEMS have been added to the paper.